# LEARNING AXIOMS TO COMPUTE VERIFIABLE SYMBOLIC EXPRESSION EQUIVALENCE PROOFS USING GRAPH-TO-SEQUENCE NETWORKS

## ABSTRACT

We target the problem of proving semantic equivalence between two complex expressions represented as typed trees, and demonstrate our system on expressions from a rich multi-type symbolic language for linear algebra. We propose the first graph-to-sequence deep learning system to generate axiomatic proofs of equivalence between program pairs. We generate expressions which include scalars, vectors and matrices and 16 distinct operators combining them, with 147 distinct axioms of equivalence. We study the robustness of the system to generate proofs of increasing length, demonstrating how incremental graph-to-sequence networks can learn to represent complex and verifiable symbolic reasoning. It achieves 93% average true positive coverage on 10,000 test cases while ensuring zero false positives by design.

## 1 INTRODUCTION

Deep neural networks have excelled at a variety of classification and reinforcement learning tasks (Goodfellow et al., 2016). However, their stochastic nature tends to hinder their ability to learn representations useful for manipulating symbolic information. For applications requiring a guarantee of correctness, such as occur in relation to determining the semantic equivalence between two programs (or symbolic expressions), a system that produces provably correct output must be developed.

In this work we target the problem of automatically computing whether two input programs are semantically equivalent (Kaplan, 1969), under a well-defined axiomatic system for equivalence using semantics-preserving rewrite rules (Dershowitz, 1985). Program equivalence is summarized as determining whether two programs would always produce the same outputs for all possible inputs, and is a central problem in computing (Kaplan, 1969; Godlin & Strichman, 2008; Verdoolaege et al., 2009). The problem ranges from undecidable, e.g. Goldblatt & Jackson (2012), to trivial in cases of testing the equivalence of a program with itself. Our work directly studies the subset of programs represented by symbolic linear algebra expressions which include scalar, vector, and matrix types for both constants and variables, and 16 different operators with 147 distinct axioms of equivalence, creating a foundation for future work with broader program semantics. For example, the expression using matrices, scalars, and a vector: $(A + B)I((a + (b - b))/a)\vec{v} - A\vec{v}$ can be proven equivalent to $B\vec{v}$ by applying 10 axioms in sequence; our work generates the proof steps between these expressions.

While prior work has shown promise for deep networks to compute some forms of program equivalence (Xu et al., 2017; Alon et al., 2019; Rabin et al., 2020), the system typically outputs only a probability of equivalence, without any reasoning or insight that can be verified easily: false positives can be produced. Programs can be represented as a tree (or graph) of symbols, and deep networks for symbolic reasoning have been studied, e.g. to compute the derivative of a symbolic expression (Lample & Charton, 2020), but such systems can produce output which is not directly explained by the system itself. In this work, we take a fundamentally different approach to the problem of symbolic program reasoning with deep networks: we make the system produce the sequence of steps that lead to rewriting one program into another, that is the *reasoning* for (or proof of) equivalence between the two programs, instead of producing directly the result of this reasoning (e.g., the transformed expression (Lample & Charton, 2020)).

We propose a method for generating training samples using probabilistic applications of production rules within a formal grammar, and then develop a graph-to-sequence (Li et al., 2016; Beck et al., 2018) neural network system for program equivalence, trained to learn and combine rewrite rules to rewrite one program into another. It can *deterministically* prove equivalence, entirely avoids false positives, and quickly invalidates incorrect answers produced by the network (no deterministic answer is provided in this case, only a probability of non-equivalence). In a nutshell, we develop the first graph-to-sequence neural network system to accelerate the search in the space of possible combinations of transformation rules (i.e., axioms of equivalence in the input language) to make two graphs representing programs structurally identical without violating their original semantics. We make the following contributions:

(i) We propose a machine learning system for program equivalence which ensures correctness for all non-equivalent programs input, and a deterministically checkable output for equivalent programs (no false positives, specificity = 100%).

(ii) We introduce `pe-graph2axiom`, the first incremental graph-to-sequence neural network system targeting program equivalence to the best of our knowledge. We provide the first implementation of such graph-to-sequence systems in the popular OpenNMT-py framework (Klein et al., 2017).

(iii) We present a complete implementation of our system operating on a rich language for multi-type linear algebra expressions. Our system provides a correct rewrite rule sequence between two equivalent programs for 93% of the 10,000 test cases. The correctness of the rewrite rule is deterministically checkable in all cases in negligible time.

## 2 PROGRAM EQUIVALENCE USING DEEP LEARNING

We outline below the key intuitions and concepts for our program equivalence system. Extensive details, including a full formal framework for program equivalence reasoning, are presented in supplementary material, Secs. A-B. We illustrate below four very simple computations, represented as graphs, that are all equivalent under various axioms of natural arithmetic.

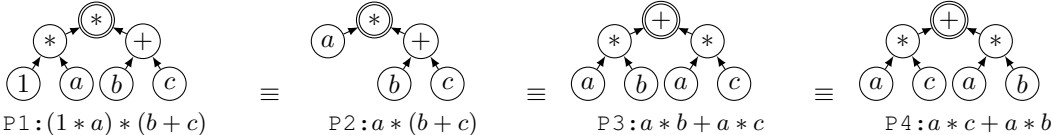

**Rewrite rules as axioms of equivalence** In this work we represent programs with symbolic expressions made of variables (e.g., $a$, $b$, $c$), operators (e.g., $+$, $*$) and neutral/absorbing elements (e.g., 1). We consider a rich linear algebra expression language, supporting three variable types (scalars as shown in P1-P4, vectors, and matrices) and 5 different variables per type; 16 operators including operators mixing different variable types such as vector-matrix product. We represent these programs as dataflow graphs Buck & Lee (1993) with a root node that is to compute a single value.

P1 is equivalent to P2 if we consider the axiom $A1 : 1_{\mathbb{N}} * x = x, \forall x \in \mathbb{N}$. This axiom is also a clear rewrite rule: the LHS expression $1_{\mathbb{N}} * x$ (with $x \in \mathbb{N}$) can be matched and replaced by the RHS expression $x$ anywhere in the program without altering its semantics. An axiom, or equivalently here a graph rewrite rule, may be applied repeatedly to different subtrees. When applying $A1$ on a specific location, the node $a$ of $P1$, we obtain an equivalent and yet syntactically different program, we note $P1 \equiv A1(a, P1)$. To assess the validity of a transformation sequence $S$ where $P2 = S(P1)$, one simply needs to check for $S$, in sequence, that each axiom is applicable at that program point, apply it to obtain a new temporary program, and repeat the process for each axiom in the complete sequence. If the sequence is verified to be valid, and $S(P1)$ is structurally equivalent to $P2$, then we have proved $P1 \equiv P2$, and $S$ forms the complete proof of equivalence between the two programs. These equivalences can be composed, incrementally, to form a complex transformation. Using $A2 : x * (y + z) = x * y + x * z, \forall x, y, z \in \mathbb{N}$ and $A3 : x + y = y + x, \forall x, y \in \mathbb{N}$, we have $P1 \equiv P4 \equiv A3(+, A2(*, A1(a, P1)))$, a verifiable proof of equivalence under our axioms between the programs $1a(b + c)$ and $ac + ab$, which involved structural changes including node deletion, creation and edge modification. Note the non-unicity of a sequence: by possibly many ways a program can be rewritten into another one, for example the sequence $P4 \equiv A2(*, A3(+, A1(a, P1)))$ also correctly rewrites $P1$ into $P4$. Conversely, a sequence may not exist: for example no sequence of the 3 above axioms allows rewriting $a + b$ into $a * b$. We call these non-equivalent in our system, that is precisely if there is no sequence of axioms that can be applied to rewrite one program into the other.

Our approach aims to compute some $S$ for a pair of programs $P1, P2$, so that $S$ is verified correct when $P1 \equiv P2$. Consequently, if $P1 \not\equiv P2$, no sequence $S$ produced can be verified correct: true negatives are trivially detected.

**Pathfinding program equivalence proofs** Intuitively, we can view the solution space as a graph, where every possible syntactically different program in the language is represented by its own vertex $v_i$. And $\exists\, e^{(A_k, x)} : v_i \to v_j$ iff $\exists A_k$ an axiom and $x \in v_i$ such that $v_j = A_k(x, v_i)$. Any two programs connected by a path in this graph are therefore semantically equivalent. Building $S$ for $P1 \equiv S(P2)$ amounts to exposing one path between $P1$ and $P2$ in this graph when it exists, the path forming the proof of equivalence. We build a deep learning graph-to-sequence system to learn a stochastic approximation of an iterative algorithm to construct such feasible path when possible, trained only by randomly sampling pairs of programs and one carefully labeled path between them. This avoids the need to craft smart exploration heuristics to make this path-finding problem practical.

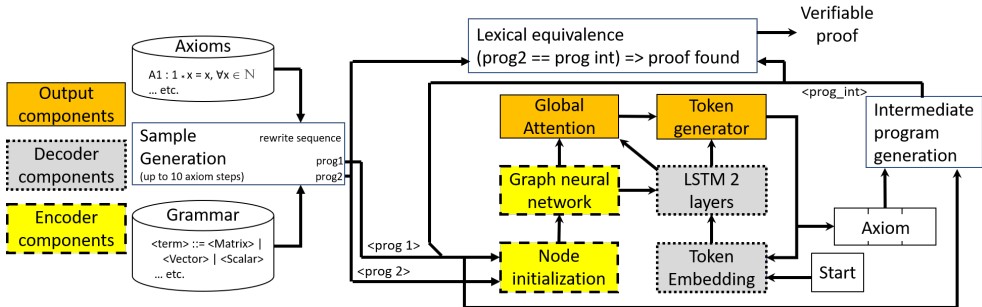

Figure 1: `pe-graph2axiom` System Overview

**Graph-to-sequence network for pathfinding** The system in Fig. 1 takes as input two programs represented as symbolic trees, and produces a sequence of axioms along with their position of application (or node) that can be used to rewrite sequentially one input program into the other input program. *Node initialization* initializes the graph neural network, converting the input programs text (e.g., $(a + (b + c))$ into nodes and edges in the *Graph Neural Network* Scarselli et al. (2009); Xu et al. (2017). The details of the network are covered in Sec. 5. In a nutshell, the key principle is to combine a memory-based neural network approach, e.g., using Long-Short Term Memory (LSTM) Hochreiter & Schmidhuber (1997) neurons and a graph neural network design (which uses Gated Recurrent Units (GRUs) internally) Beck et al. (2018) that matches our program graph representation. *Token embedding* is a neural network layer in which tokens are assigned a learnable multidimensional embedding vector Mikolov et al. (2013). Each layer in *LSTM 2 layers* has 256 neurons, which support sequence generation. *Token generator* is the final output portion of the network. It learns to output the tokens based on the current LSTM hidden states and the *Global Attention* from the graph neural network. As each token is output, it feeds back into the LSTM layer through the embedding layer to affect its next state. We use a sequence generation principle, using a global attention mechanism Luong et al. (2015) to allow observation of program graph node information while generating the axiom and location on which it is applied. We specifically study the robustness of our approach to generate proofs of increasingly complex length, contrasting models to output the entire path at once with `pe-graph2axiom` which incrementally builds the sequence one step at a time in Sec. 6.

## 3 RELATED WORK

**Program analysis with machine learning** Numerous prior works have employed (deep) machine learning for program analysis (Allamanis et al., 2018a; Alon et al., 2019; Tufano et al., 2019; Lacomis et al., 2019; Raychev et al., 2015; Bavishi et al., 2017). PHOG (Bielik et al., 2016) presents a probabilistic grammar to predict node types in an AST. Program repair approaches, (Tufano et al., 2019; Chen et al., 2019) are deployed to automatically repair bugs in a program. Wang et al. (2018) learn to extract the rules for Tomita grammars (Tomita, 1982) with recurrent neural networks. The learned network weights are processed to create a verifiable deterministic finite automata representation of the learned grammar. This work demonstrates that deterministic grammars can be learned with RNNs, which we rely on. Recent work by Rabin et al. (2020) shows that GGNNs learn more general representations of semantically equivalent programs than code2vec (Alon et al., 2019), which creates code representations using AST paths. Bui (2020) shows that using semantics

preserving transformation can improve machine learning on code; we target learning to identify the transformations that prove equivalence.

**Symbolic mathematics using machine learning** has received recent interest. Fawzi et al. (2019) develop a system which learns to apply a set of inequality axioms and derived lemmas using a reinforcement learning model with a feed-forward neural network. A challenge of using reinforcement learning is the determination of a feasible reward function from the environment and the resulting training time. Bansal et al. (2019) present a system that can interact with a large rule set to score tactics in the search for a proof but discuss the compute time limitations of training a reinforcement learning network for theorem proving. Unlike their model, we output both the tactic (axiom) as well as the location to apply it (eliminating the need to score various premises individually). Paliwal et al. (2019) explores using graph neural networks as applied to theorem proving in the HOList environment. Our work aims at laying the foundation for program equivalence proofs by studying a language subset that includes variable types and maintains a high accuracy. Our approach to data modeling is similar to Lample & Charton (2020), who randomly create representative symbolic equations to develop a deep learning sequence-to-sequence transformer model which can perform symbolic integration. They note that their system requires an external framework to guarantee validity, however our system outputs a verifiable reasoning that is straightforward to check for correctness, guaranteeing no false positive and the correct handling of all true negatives.

**Graph neural networks** Graph neural networks (Scarselli et al., 2009; Wu et al., 2019), such as a graph-to-sequence network with attention, have been investigated for natural language processing (Beck et al., 2018). Structure2vec (Xu et al., 2017) uses a graph neural network to detect binary code similarity where a probability of equivalence is sufficient. Allamanis et al. (2017) explore methods in which semantically exquivalence expressions have similar embedding representations. Allamanis et al. (2018b) use graph neural networks to analyze code sequences and add dedicated edge types to improve the system's ability to reason about the code. Their work achieves 84% accuracy on correcting variable misuse cases and provides insights to useful edge types. The G2SKGE model (Li et al., 2019) uses a node embedding (information fusion mechanism) to capture node relationships similar to ours. Given the inherent graph structure of expressions we manipulate, and potential advantages of using graph neural networks for symbolic computing (Lamb et al., 2020; Rabin et al., 2020), our model makes use of a graph-to-sequence model for producing axiomatic proofs.

**Static program equivalence** Algorithms restricted to specific classes of programs have been developed (Verdoolaege et al., 2012; Alias & Barthou, 2004; Barthou et al., 2002; Iooss et al., 2014). These approaches typically restrict to proving the equivalence of different schedules of the operations, possibly via abstract interpretation (Schordan et al., 2014; Churchill et al., 2019) or even dynamically (Bao et al., 2016). The problem of program equivalence we target may be solved by other (smart) brute-force approaches, where a problem is solved by pathfinding. This includes theorem provers (Bertot & Castéran, 2013; Paulson), which handle inference of axiomatic proofs. Program rewrite systems have been heavily investigated, (Dershowitz, 1985; Steffen, 1991; Clarke et al., 2003; Visser et al., 2003; Namjoshi & Kurshan, 2000; Kalvala et al., 2009; Mansky & Gunter, 2010). Our contribution is not in the formal definition of program equivalence we presented, semantics-preserving rewrite systems have been studied (Visser, 2004; Lucanu & Rusu, 2015; Reddy, 1989). But recognizing this compositional formalism is well suited to deep learning graph-to-sequence systems was essential. The merits of stochastic search to accelerate such systems has been demonstrated (Murawski & Ouaknine, 2005; Hérault et al., 2004; Gogate & Domingos, 2012). The novelty of our work is to develop carefully crafted graph-to-sequence neural networks to automatically learn an efficient pathfinding heuristic for this problem.

## 4 SAMPLES GENERATION

The careful creation of our training dataset is key: as we let the DNN learn *by example only* what the axioms are and when they are applicable in the structure of a program, we must carefully sample the space of equivalences to ensure appropriate distributions of the examples. We produce a final dataset of tuples $(P1, P2, S)$, a pair of input programs and a possible rewrite rule sequence that proves the pair equivalent. Duplicates are removed such that all samples have a unique $P1$. From this dataset, we create 1,000,000 training samples, 10,000 validation samples, and 10,000 test samples. We outline below its generation principles; extensive details and the algorithms used are presented in section C.1.

**Random sample generation** Deep learning typically requires large training sets to be effectively deployed, hence we developed a process to automate sample generation. We specifically use randomized program generation algorithms that are inspired by a given language grammar. By randomly choosing between production rulse, one can build random parse trees by simply iterating the grammar. The leaves obtained will form a sentence accepted by the language, i.e., a program Bielik et al. (2016). We limit to programs of 50 nodes in the program tree (or AST), with a maximal tree depth of 7. We assert that our random production rule procedure has a non-zero probability of producing any program allowed by the grammar for our datasets.

We produce equivalent program samples by pseudo-randomly applying axioms on one randomly generated program to produce a rewrite sequence and the associated equivalent program. Given a randomly selected node in the program graph, our process checks which axiom(s) can be applied. E.g., the $+_m$ operator may have the Commute axiom category applied, or it may have the Transpose axiom category applied, which affects the operator's children.

**Final experimental dataset: AxiomStep10** To train our network to produce one axiom step at a time, as described in Sec. 2, AxiomStep10 has a single axiom in each output sequence $S$. For a complete proof $S : A_1(A_2(...)$ in a $(P1, P2, S)$ we generated made of N axioms, we then create N training examples for the network: $(P1, P2, A_N)$ the first intermediate step by applying the first axiom, then $(A_N(P1), P2, A_{N-1})$, etc. We limit proof length to 10 axioms in our experiments (hence AxiomStep10). Test samples only have the original and target program and the network proposes axioms which create intermediate programs towards the proof, fed back to the system.

Table 1: Distribution for the 14 axiom categories in AxiomStep10 test set. Considering scalars (a, b, ...), vectors ($\vec{v},\vec{w}$, ...) and matrices (A, B, ...) types combinations, 147 distinct axioms are represented.

| Axiom Category | Example axiom(s) | Samples with | Axiom Category | Example axiom(s) | Samples with |
|---|---|---|---|---|---|
| Cancel | (A-A)→O,(b/b)→1 | 13.8% | NeutralOp | $(\vec{v} - \vec{o}) \rightarrow \vec{v}$ | 40.0% |
| DoubleOp | $A^{t^t} \rightarrow A$, 1/1/x→x | 7.3% | AbsorbOp | (A*O)→O, (b*0)→0 | 30.3% |
| Commute | $(a + b) \rightarrow (b + a)$ | 48.6% | DistributeLeft | (a + b)c → ac + bc | 36.3% |
| DistributeRight | a(b + c) → ab + ac | 27.8% | FactorLeft | ab + ac → a(b+c) | 6.1% |
| FactorRight | ac + bc → (a+b)c | 9.0% | AssociativeLeft | a(bc) → (ab)c | 46.3% |
| AssociativeRight | (ab)c → a(bc) | 43.1% | FlipLeft | -($\vec{v}$ - $\vec{w}$) → $\vec{w} - \vec{v}$ | 8.4% |
| FlipRight | a/(b/c) → a(c/b) | 26.1% | Transpose | $(AB)^t \rightarrow B^t A^t$, | 11.1% |

**Datasets to study generalizability and robustness** In order to study our model's ability to generalize, we have created alternate datasets on which to train and test models which are summarized in table 2. *WholeProof10* will help us contrast learning approaches. This dataset has the complete proof sequence $S$ made of $N \geq 1$ axioms as reference output for a program pair, while for AxiomStep10, $N = 1$. Models trained on WholeProofX must maintain internal state representing the graph transformations that the axioms create. They are not "iterative": a single inference is expected to produce the complete proof; in contrast to AxiomStep10 for which a single axiom of the sequence is produced at each inference step. Training long output sequences can benefit from complex training approaches such as Professor forcing Lamb et al. (2016), but we will show that our AxiomStep10 model generalizes well with our sequence model training approach.

Table 2: Datasets used for studies in experiments.

| Dataset | AST depth | AST #nodes | Proof length | Iterative | Comments |
|---|---|---|---|---|---|
| AxiomStep10 | 2-7 | 2-50 | 1-10 | Yes | Main dataset - most general |
| AxiomStep5 | 2-6 | 2-25 | 1-5 | Yes | Less general version of AxiomStep10 |
| WholeProof10 | 2-7 | 2-50 | 1-10 | No | Constrains AST traversal |
| WholeProof5 | 2-6 | 2-25 | 1-5 | No | Less general version of WholeProof10 |

**Complexity of equivalence space** Figure 2 provides a view of the complexity of the equivalence problem we tackle. The distribution of the dataset per proof length is displayed in the right chart; the left chart shows by size of bubble the number of test samples with a given number of *semantics-preserving* axioms that may be implemented as the first step of the proof and the proof length needed. There is a large number of proofs possible in our system, as detailed in Appendix C.3. For example, for proofs of length 5, about 340,000 proofs made only of legal applications of axioms can be performed on the average sample in our dataset. Since many programs have multiple possible proofs, about 10,000 different programs can be produced, only one of which is the target to prove, i.e.,

Figure 2: Distribution of axiom possibilities and proof complexity for test datasets.

randomly drawing a valid 5 axiom proof on a program known to be 5 axiom steps from the target has roughly a 1 in 10,000 chance of being a correct proof of equivalence between the two programs.

## 5 PE-GRAPH2AXIOM: DNN FOR PROGRAM EQUIVALENCE

Fig. 1 overviews the entire system architecture including sample generation, the `pe-graph2axiom` network, and the rewrite checker. Key design decisions are presented below.

**Graph neural network** The sample generation discussed in section 4 provides input to the Node Initialization module in Fig. 1 to create the initial state of our graph neural network (Beck et al., 2018). For each node in the program graph, a node will be initialized in our graph neural network with a value that encodes the AST level and language token of the program node. To interconnect the edges we support 9 edge types and their reverse edges which allows information to move in any direction necessary: 1) left child of binary op, 2) right child of binary op, 3) child of unary op, 4) root node to program 1, 5) root node to program 2, 6-9) there are 4 edge types for the node grandchildren (LL, LR, RL, RR). The node states and edge adjacency matrix represent the initial graph neural network state.

After initialization, the graph neural network iterates 10 times in order to convert the initial node state into the embeddings needed for rewrite rule generation. Given an initial hidden state for node $n$ of $x_n(0)$, $x_n(t+1)$ is computed with a learnable function $f$ which combines the current hidden state $x_n(0)$, the edge types $l_{in[n]}$ of edges entering node $n$, the edge types $l_{out[n]}$ of edges exiting node $n$, and the hidden states $x_{ne[n]}$ of the neighbors of node $n$: $x_n(t+1) = f(x_n(t), l_{in[n]}, x_{ne[n]}(t), l_{out[n]})$.

Each edge type has a different weight matrix for learning, allowing aggregation of information into a given node related to its function in the full graph of the program. The root node's initial state along with the edge types connecting it to the program graph trees allow it to aggregate and transfer specific information regarding rewrite rules as demonstrated by our experimental results. This is a novel feature of our network not used in prior work with GNNs on program analysis (Allamanis et al., 2018b; Xu et al., 2017).

**Graph neural network output to decoder** After stepping the GGNN, the final node values are used by the decoder in two ways to create rewrite rules. First, the final root node value $x_{root}(10)$ is fed through a learnable bridge function to initialize the LSTMs of the decoder. In this way, the aggregated information of the 2 programs seeds the generation of rewrite rules. The LSTMs update as each output token $y_j$ is generated with a learnable function based on the current decoder hidden state $h_j^d$ at decoder step $j$ and the previous output token $y_{j-1}$ (Chen et al., 2019). Second, all nodes in the graph can be used by an attention layer (Bahdanau et al., 2015). The attention layer creates a context vector $c_j$ which can be used by a learnable function $g$ when computing the probability for generating the $j$th output token $P(y_j)$: $P(y_j \mid y_{j-1}, y_{j-2}, ..., y_0, c_j) = g(h_j^d, y_{j-1}, c_j)$. Because `pe-graph2axiom` has a robust output verification, we make use of beam search to track up to 10 likely candidates for proofs of equivalence.

By using the root node only for seeding the initial hidden state $h_0^d$ of the decoder, the weights associated with its connections to the program graphs for $P1$ and $P2$ learn to represent the information necessary for the rewrite rule sequence. In parallel, after the graph neural network iterations complete, the final embedding for all the nodes in the graphs for $P1$ and $P2$ are only used by the attention network, so their final embedding represents information useful during rewrite rule generation.

**Intermediate program generation** `pe-graph2axiom` applies the axiom and program node chosen by the neural network token generator to the input program to create an intermediate program

$P'$ on the path from $P1$ to $P2$. If this program is equal to $P2$, then our axiom path is complete, otherwise the new pair $P', P2$ is inferred to determine the next axiom step.

**Incremental versus non-incremental sequence production** The models we train on the Axiom-Step5, WholeProof10, and WholeProof5 datasets have the same neural network hyperparemeters as the AxiomStep10 data model. However, the models for WholeProof10 and WholeProof5 are trained to output the entire sequence of axioms needed to prove the 2 programs identical, hence these models do not make use of the intermediate program generation and instead have a component which checks whether the full sequence of axioms legally transforms $P1$ into $P2$. We encode the path to the AST node an which to apply an axiom using 'left' and 'right' tokens which specify the path from the current program root node. This encoding is sufficient for the iterative model and necessary to allow the non-iterative model to identify nodes which may not have been in the initial AST for $P1$. The non-iterative models must learn a representation in the LSTM network to allow them to track AST transformations as they are generated.

## 6 Experimental Results

We focus our experiments below on 4 key questions: 1) Is performance related to input program size? 2) Is performance related to proof length? 3) Is the incremental, per-axiom approach more generalizable than producing the full sequence in a single inference step? And 4) Is performance consistent across a range of datasets, including human-written examples?

**Implementation setup** We developed the neural network system presented in the OpenNMT-py system (Klein et al., 2017), adding on a new encoder based on a prior implementation of gated graph neural networks (Li et al., 2016). For our training and evaluation experiments, we use systems with Intel Xeon 3.6GHz CPUs and 6GB GeForce GTX 1060 GPUs. During training, we save a model snapshot every 50,000 iterations and score the accuracy the model achieved on the validation dataset. Graphs showing that validation accuracy plateaus at 200,000 to 300,000 iterations are provided in section F. We run each model twice and evaluate the test set using the saved model which achieved the highest validation score.

**Evaluation procedure and neural network alternatives** The benefits of key components of our neural network model are studied in table 3. The bidirectional RNN model is similar to state-of-the-art sequence-to-sequence models used for program repair (Chen et al., 2019). The results for the graph-to-sequence model without attention show the benefit of providing the node information during the axiom generation process.

Table 3: `pe-graph2axiom` mini ablation study.

| Model description | Beam width | | | |
|---|---|---|---|---|
| | 1 | 2 | 5 | 10 |
| Bidirectional RNN seq-to-seq with attention | 48 | 62 | 71 | 75 |
| Graph-to-sequence w/o attention | 73 | 81 | 87 | 90 |
| `pe-graph2axiom` model | 76 | 84 | 90 | **93** |

Our final design was influenced by explorations we performed on varied models, datasets, and hyperparameters such as LSTM layers and graph neural network parameters. In relation to the model's ability to learn a representation of the proof sequence, we note that our GGNN initialization using the root node connection to the decoder outperforms the embedding learned by a bidirectional RNN model. Also, we found that averaging the embedding of all graph nodes had about 10% lower accuracy than using the more specific root node information. Numerous additional results are reported in Suppl. material F.

**Generalizing across different datasets** We specifically look at the generalization potential for our models by studying their success rate as a function of the input program complexity, represented as the AST depth, in Table 4, and as a function of the output complexity, represented by the proof length in Table 5, all using a beam size of 10. We designed our datasets in Sec. 4 to study how well `pe-graph2axiom` generalizes and to assess we are not overfitting on training data. Extensive in-depth additional experimental results are presented in Suppl. Material F, we summarize key results only below.

Table 4: Performance vs. AST size: counts and percentage pass rates.

| AST depth | Testset Sample Count | | Model trained on AxiomStep5 | | Model trained on AxiomStep10 | |
|---|---|---|---|---|---|---|
| | AS5 | AS10 | AS5 | AS10 | AS5 | AS10 |
| 2-6 | 10000 | 6865 | 99 | 93 | 99 | 94 |
| 7 | 0 | 3135 | n/a | 86 | n/a | 92 |
| All | 10000 | 10000 | 99 | 90 | 99 | **93** |

Table 4 illustrates the ability of a model trained on AxiomStep5 (i.e., limited to proofs of length 5) to perform well when evaluated on the more complex AxiomStep10, which includes proofs of unseen length of up to 10. The robustness to the input program complexity is illustrated with the 86% pass rate on AST depth 7, for the model trained on AxiomStep5 which never saw programs of depth 7 during training.

Table 5: Performance vs. proof length: percentage pass rates.

| Axiom Count in Proof | Model trained on WholeProof5 (WP5) | | | | Model trained on WholeProof10 (WP10) | | | | Model trained on AxiomStep5 (AS5) | | | | Model trained on AxiomStep10 (AS10) | | | |
|---|---|---|---|---|---|---|---|---|---|---|---|---|---|---|---|---|
| | WP5 | WP10 | AS5 | AS10 | WP5 | WP10 | AS5 | AS10 | WP5 | WP10 | AS5 | AS10 | WP5 | WP10 | AS5 | AS10 |
| 1-5 | 95 | 89 | 44 | 44 | 94 | 93 | 44 | 44 | 99 | 97 | 99 | 98 | 99 | 98 | 99 | **98** |
| 6 | | 14 | | 4 | | 72 | | 5 | | 81 | | 88 | | 90 | | **93** |
| 7 | | 0 | | 1 | | 63 | | 2 | | 67 | | 81 | | 83 | | **87** |
| 8 | | 0 | | 0 | | 54 | | 1 | | 54 | | 75 | | 73 | | **82** |
| 9 | | 0 | | 0 | | 47 | | 0 | | 35 | | 64 | | 63 | | **74** |
| 10 | | 0 | | 0 | | 34 | | 0 | | 24 | | 57 | | 46 | | **66** |
| All | 95 | 66 | 44 | 27 | 94 | 84 | 44 | 27 | 99 | 87 | 99 | 90 | 99 | 93 | 99 | **93** |

Table 5 compares the results of our 4 models, each trained on one of our 4 datasets, and evaluated with the test set of all 4 datasets. The models all have identical hypermeter settings. We observe the inability of models trained to output the whole proof to generalize to proofs of higher length (WP5 model on AS10/WP10), with near zero success rate. However, per-axiom models (AS5 and AS10) show potential for generalization to proof length: AS5 model performs well when evaluated on AS10, showing the ability to produce proofs of length/complexity unseen in training. Overall, the success rate degrades gracefully with proof length, bottoming at 66% for AS10 for proofs of length 10.

**Human written test expressions from Khan academy exercises** Unfortunately there is a dearth of existing large reference datasets for equivalence of linear algebra expressions, which justified our careful dataset creation approach in Sec. 4 and their upcoming public release. However numerous math exercises involve exactly this problem, and can provide small but human-written datasets. We solve all of the matrix expression equivalence programs from 2 relevant Khan academy modules designed to test student's knowledge of matrix algebra (Khan, 2020). Our AxiomStep10 model is able to correctly prove all 15 equivalent pairs from the modules with beam width 1 and wider. With a beam width of 10, the WholeProof10 model proved 12. An example problem solvable by AxiomStep10 but not WholeProof10 is: $c(1A + B) = cB + cA$ which can be proven by applying the rewrite rules NeutralOp, DistributeRight, and Commute to the proper nodes. The WholeProof10 model mostly fails because it was not trained on how to apply repeated transformations at the same point in the AST. This suggests AxiomStep10 has generalized well to these hand-written problems.

## 7 CONCLUSION

In this work, we presented `pe-graph2axiom`, the first graph-to-sequence neural network system to generate verifiable axiomatic proofs (via rewrite rules) for equivalence for a class of symbolic programs. Evaluated on a rich language for linear algebra expressions, this system guarantees no false positives are ever produced, all true negatives are preserved, and produces correct proofs of up to 10 axioms in length in 93% of the truly equivalent cases. We believe the performance of our approach comes in part from using graph neural networks for what they aim to excel at: learning efficient heuristics to quickly find paths in a graph; and the observation that program equivalence can be cast as a path-based solution that is efficiently found by such networks.

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

# Supplementary Materials:
## Learning Axioms to Compute Verifiable
## Symbolic Expression Equivalence Proofs
## using Graph-to-Sequence Networks

### DOCUMENT OVERVIEW

This document supplements the submission *Learning Axioms to Compute Verifiable Symbolic Expression Equivalence Proofs using Graph-to-Sequence Networks*. We have provided below numerous additional information for completeness. We also provide access to anonymized software artifacts to replicate our results. Our supplementary materials are organized as follows:

- Appendix A of this document revisits the motivation and overview of our system in more depth.
- Appendix B of this document formalizes the complete framework for axiomatic program equivalence we leverage in our work.
- Appendix C of this document presents the dataset generation approach we developed.
- Submitted with this supplementary document is the `pe-graph2axiom` directory which contains our code for generating datasets and training models, the testsets for our 4 key datasets, and key results files from our testset evaluations. The test dataset for AS10, our most complex dataset, is provided in pe-graph2axiom/data/AxiomStep10/all_test_fullaxioms.txt for convenience. The results file showing 9,310 successful proofs found on our 10,000 sample testset for AS10 is pe-graph2axiom/runs/AxiomStep10/mbest_300_AxiomStep10/search10.txt.
- Appendix D of this document presents exhaustively the language for complex linear algebra expressions we evalaute on, including the list of all 147 axioms of equivalence we learned.
- Appendix E of this document presents additional details about the neural network architectures we developed.
- The anonymized url `https://gofile.io/d/gTPmaz` contains all trained models evaluated in this paper, including scripts to train them directly from our datasets, using OpenNMT. The AS10 model, our best/golden model, is also provided in file pe-graph2axiom-big/runs/AxiomStep10/final-model_step_300000.pt of this archive for convenience.
- Appendix F of this document presents complementary experimental results and additional in-depth details on results presented in the main paper body.

## A    MOTIVATION, INTUITIONS AND SYSTEM OVERVIEW

### A.1    INTUITIONS ON AXIOMATIC PROGRAM EQUIVALENCE

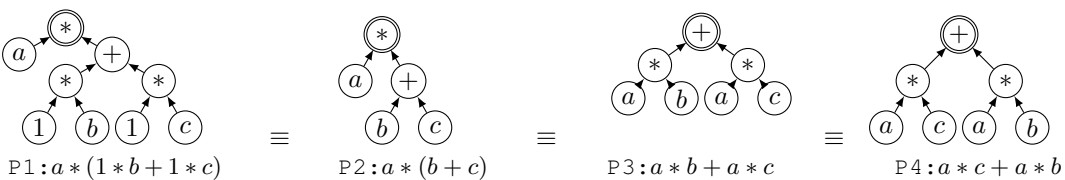

Figure 3: Examples of Computations

**Input program representation**    We illustrate in Fig. 3 four very simple computations, represented as graphs, that are all equivalent under various axioms of natural arithmetic. For example, `P1` models the expression $a(1b + 1c)$, one can imagine it to be the result of $a(db + dc)$ after e.g. constant-propagation of 1 to $d$ by a compiler. They are defined by a single root, have nodes which can be operations consuming the value of their immediate predecessor or terminal/input values, and a node produces a value that can be used by its immediate successors. In essence this is a classical dataflow representation of the computation Buck & Lee (1993), and what our system uses as input program representation.

In this work we represent programs with symbolic expressions made of variables (e.g., $a$, $b$, $c$), operators (e.g., $+$, $*$) and neutral/absorbing elements (e.g., 1). We consider a rich linear algebra expression language, supporting three variable types (scalars as shown in `P1`-`P4`, vectors, and

matrices) and 5 different variables per type; 16 operators including operators mixing different variable types such as vector-matrix product. Details are provided in Sec. B and Sec. D.

**Rewrite rules as axioms of equivalence**   Consider the programs P1 versus P2. The multiplication of an integer value by 1 does not change the value, if we rely on an axiom of equivalence $A1 : 1_\mathbb{N} * x = x, \forall a \in \mathbb{N}$. This axiom specifies a strict criterion of application: the node must be of type $\mathbb{N}$, the expression pattern must be $1_\mathbb{N} * x$; and a strict rewrite rule: replace a sub-graph $1_\mathbb{N} * x$ for any $x \in \mathbb{N}$ by the graph $x$. In other words, replacing $1 * b$ by $b$ in P1 is a semantics-preserving rewrite, from the axiom of equivalence $A1$. In this work we view the problem of program equivalence as finding a sequence of semantics-preserving rewrites, each from a precisely defined axiom of equivalence, that rewrites one program into the other. If one program can be rewritten by a sequence of individually-correct semantics-preserving transformations into another one, then not only are they equivalent under the set of axioms used, but the sequence forms the constructive and verifiable proof of equivalence.

**An example**   In this work we illustrate and experimentally evaluate our system using a rich linear algebra expression language because it exposes clearly (and intuitively) the various key concepts that must be handled: (1) operating on dataflow graphs as input, supporting transformations that can (2) delete or (3) create new nodes in the graph, and transformations that (4) manipulate entire subtrees. We also wanted a language with (5) multiple variable types, e.g. scalars, vectors and matrices and (6) a large number of different operators with (7) distinct axioms applicable for each. All of these are captured in the language we experiment with, see Sec. B for its formal definition.

When applying the axiom $A1 : 1 * x = x, \forall x \in \mathbb{N}$ on the program P1 for its node $b$, we obtain an equivalent and yet syntactically different program, we have $P1 \equiv A1(b, P1)$. Applying the same axiom $A1$ on $c$ in the resulting program leads to program P2, and $P2 \equiv P1 \equiv A1(c, A1(b, P1))$.

Consider now the axiom $A2 : x * (y + z) = x * y + x * z, \forall x, y, z \in \mathbb{N}$. This is the standard distributivity axiom on natural arithmetic. In terms of graph transformations, this is a complex rewrite: a new node is created ($*$), one node is moved ($+$ to the root), and edges are significantly modified. When this complex, but semantics-preserving, rewrite is applied to P2, we obtain P3, that is $P3 \equiv A2(*, A1(c, A1(b, P1)))$.

Finally consider the axiom $A3 : x + y = y + x, \forall x, y \in \mathbb{N}$, the standard commutativity axiom for $+$. The graph transformation does not change the number of nodes nor edges, instead only alters two specific edges. Note that as the previous axioms, it also illustrates operations on sub-graphs: indeed $x$ and $y$ do not need to be input/terminal nodes, they can be any subgraph producing a value of the proper type. This is illustrated by applying it on P3 to obtain P4, that is the computation $ac + ab$. We have $P4 \equiv A3(+, A2(*, A1(c, A1(b, P1))))$, a verifiable proof of equivalence under our axioms between the programs $a(1b + 1c)$ and $ac + ab$, which involved structural changes including node deletion, creation and edge modification. Note the bidirectional nature of the process: one can rewrite from $a(1b + 1c)$ to $ac + ab$, or the converse using the same (but reverted) sequence. Note also the non-unicity of a sequence: by possibly many ways a program can be rewritten into another one, for example the sequence $P4 \equiv A3(+, A1(c, A1(b, A2(*, P1)))$ also correctly rewrites P1 into P4. Conversely, a sequence may not exist: for example no sequence of the 3 above axioms allow to rewrite $a + b$ into $a * b$. We call these non-equivalent in our system, that is precisely if there is no sequence of axioms that can be applied to rewrite one program into the other.

**The need for a verifiable procedure**   A key motivation of our work is to enable in a safe and provably correct way the use of machine learning for program equivalence by ensuring no false negative can be produced. For full automation of the process, we focus on ensuring correctness in case an equivalence result is computed by the system. That is, our system by design answers only with a probability of confidence that the two programs are not equivalent, but *it produces a verifiable procedure to assess equivalence* otherwise. We believe such an approach is key for a practical, automated deployment of neural networks for program equivalence: verifiably proving equivalence to ensure no false positive, while tolerating a moderate amount of false negative (i.e., missing that two programs were in fact equivalent).

Applications of such a system include for example the automatic generation and correction of exercises for students, where they typically need to prove equivalence between two formulas by successive

application of other formulas/axioms. Languages like e.g. Matlab could use interactive checking of the equivalence between the expression being typed and the pre-existing library implementations (e.g., BLAS-based Goto & Van De Geijn (2008)) to use instead accelerated implementations when possible in real-time. But we have designed and evaluated our system in a robust enough way to be applicable to a wide variety of languages and problems, as long as they can be cast in the framework in Sec. B.

**The space of equivalences**    Intuitively, our approach to program equivalence is as follows. We can intellectually reason on a graph for equivalent programs where each node represents a distinct program in the language, and two nodes (i.e., two different programs) are connected by a directed edge iff the source node can be rewritten as the target node by the application of a single one of the pre-defined axioms for equivalence. The edge is labeled by the axiom used and the specific position in the source node's program to where it needs to be applied to obtain the program in the target node. Then there will be one or more paths in this graph from the two nodes modeling the two input programs if they are equivalent (one can be rewritten into the other while preserving semantics); and no path if no such rewrite is possible, that is the programs would be not equivalent in our framework. Exposing a path between two nodes is sufficient to prove the equivalence of their associated programs.

This path is exactly a sequence of rewrite rules from one program to another. To test the correctness of an arbitrary sequence, i.e., verify if this path exists in the graph and assess equivalence if it does, one then needs to simply apply the proposed sequence to one of the input programs: verify at each step that the rewrite in the sequence is indeed applicable (by a simple check of the applicability of the axiom at this particular program point), and eventually ensure the rewritten program is identical to the other input one. This test can be computed in time mostly linear with the program size in our framework, and when successful it implements a constructive proof of equivalence between the two programs.

**Pathfinding equivalence proofs**    When formulating the program equivalence problem this way, we can then view its solution as learning how to build at least one feasible path between any two pairs of nodes in the above graph, when it can exist. We can see that by design, there is a lot of redundancy in this space: the same labeled path will occur between many different pairs of programs (e.g., those where only the variable symbols differ), and there are typically many paths between the same two (equivalent) programs. This creates opportunities for the system to learn program representation and path construction techniques more easily.

Our key contribution is the development of a deep learning framework that learns this procedure automatically. The neural network system we build is trained by randomly sampling this graph, with samples made of two nodes and a path between them when training on equivalent programs, and an empty path otherwise. We specifically learn a generalization of the problem of finding paths in this graph as follows. We represent input programs in a carefully-crafted normalized dataflow-like graph encoded as a gated graph neural network Scarselli et al. (2009); Beck et al. (2018), to enable structural, size-tolerant reasoning by the network on the inputs. It is combined with a global attention-based mechanism and a memory-based LSTM Hochreiter & Schmidhuber (1997) decoder which can memorize graph changes for producing the rewrite sequence and enable path-size tolerant reasoning, while following the properties of the axioms for equivalence.

In a nutshell, we make the network learn a stochastic approximation of an iterative algorithm that would be able to construct a feasible path (when possible) between any two pairs of nodes in this equivalence graph, but trained simply by randomly sampling pairs of nodes and one carefully labeled path between them. This avoids entirely the need to craft smart exploration heuristics to make this path-finding problem feasible in practice. This is instead what we let the neural network learn automatically; and specifically why we implemented graph neural networks to solve this problem Scarselli et al. (2009); Xu et al. (2017). We rely on the network to suggest a transformation path by inference, and then verify its validity in linear time.

## A.2    SYSTEM OVERVIEW

To implement our approach, we enumerate randomly valid sentences in a language, and a set of axioms of equivalence expressible as semantics-preserving rewrite rules from one to the other. The system in Fig. 4 takes as input two programs represented as symbolic trees representing a dataflow

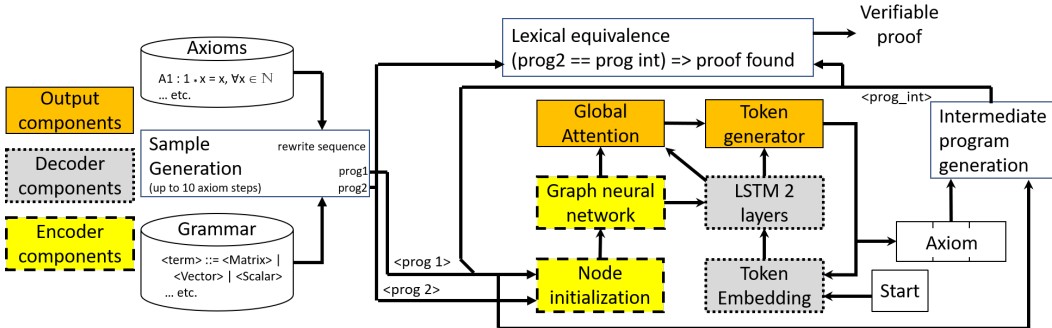

Figure 4: `pe-graph2axiom` System Overview

graph of an expression computation, and eventually produces a sequence of axioms along with their position of application (or node) that can be used to rewrite sequentially one input program into the other input program. As each axiom is produced, it is checked to insure it is a legal application within the grammar and if the transformed program matches the target then a correct proof of equivalence has been found. To train the system, we generate pairs of equivalent programs by iterating the axioms with random probability on one program, thereby generating both a path to equivalence and the target program. Random programs are generated so as to respect the grammar defined. The training set is then appropriately selected from these random samples, as detailed in Sec. E. *Node initialization* initializes the graph neural network, converting the input programs text (e.g., $(a+(b+c))$ into nodes and edges in the *Graph Neural Network* Scarselli et al. (2009); Xu et al. (2017). The details of the network are covered in Sec. 5. In a nutshell, the key principle is to combine a memory-based neural network approach, e.g., using Long-Short Term Memory (LSTM) Hochreiter & Schmidhuber (1997) neurons and a graph neural network design (which uses Gated Recurrent Units (GRUs) internally) Beck et al. (2018) that matches our program graph representation. *Token embedding* is a neural network layer in which tokens are assigned a learnable multidimensional embedding vector Mikolov et al. (2013). Each layer in *LSTM 2 layers* has 256 neurons, which support sequence generation. *Token generator* is the final output portion of the network. It learns to output the tokens based on the current LSTM hidden states and the *Global Attention* from the graph neural network. As each token is output, it feeds back into the LSTM layer through the embedding layer to affect its next state. We use a sequence generation principle, using a global attention mechanism Luong et al. (2015) to allow observation of program graph node information while generating the axiom and location on which it is applied. As developed below, we specifically study the robustness of our approach to generate proofs of increasingly complex length, contrasting models to output the entire path at once with `pe-graph2axiom` which incrementally builds the sequence one step at a time, as shown in Sec. F.

## B FORMAL FRAMEWORK FOR PROGRAM EQUIVALENCE

We now present the formalism we use in this work to represent programs and their equivalences. We carefully co-designed the chosen problem representation and the (graph) neural network approach to make the best use of machine learning via deep networks, as discussed in Sec. E.

### B.1 PROGRAM REPRESENTATION

A key design aspect is to match the capability of the neural network to model the input as a walkable graph with the actual input program representation to be handled. We therefore model programs in a dataflow-like representation (i.e., a directed graph), using a single root/output node, that is programs produce a single final value, at their root.

Note we restrict programs evaluated in our work to be directed acyclic graphs, and ensure every node has a single successor: if a value is used multiple times, nodes representing this value are replicated in the input program accordingly, as shown in Fig. 3 where several nodes modeling the same variable $a$ are used in the same program.

A program is represented by its program graph, defined as follows.

**Definition 1** (Program graph node). *A node $n \in N$ in the program graph models n-ary operations and input operands. A node produces a value which can be consumed by any of its immediate successors in the graph. When a node has no predecessor, it models an input value. The output value for the computation is produced by the unique root node $n_{root}$ of the graph, the only node without successor.*

**Definition 2** (Program graph directed edge). *A directed edge $e_{n_1,n_2} : n_1 \rightarrow n_2$ with $n_1, n_2 \in N$ in the program graph connects the producer of a value ($n_1$) to a node consuming this value in the computation.*

**Definition 3** (Program graph). *A program graph $G$ is a directed dataflow graph modeling the computation, made of nodes $n_i \in N$ and edges $e_{n_i,n_j} \in E$ as defined in Def. 1 and Def. 2. That is, $G = \langle n_{root}, N, E \rangle$. There is no dangling edge nor unconnected node in G.*

**Language of linear algebra expressions**  We developed a complex-enough language to evaluate carefully our work, that captures rich linear algebra expressions. Specifically, we support 3 types of data/variables in the program: scalars, vectors and matrices. We use the standard notation $a, \vec{a}, A$ for scalars, vectors and matrices. We evaluate using different variable names for each of the 3 types above, along with their identity and absorbing elements.

We also model a rich set of operators, mixing different unary and binary operations for each type. Specifically, we support $*_s, +_s, -_s, /_s$ between scalar operands, $+_v, -_v, *_v$ between vectors and $+_m, -_m, *_m$ for matrices. For $-, /$ we also support their unary version for all types, e.g. $^{-1_s}$ for unary scalar inversion and $-_m$ for unary matrix negation. For example $a^{-1_s}$ computes to $1/a$. We support two specific unary matrix operations, transpose $^{t_m}$ and matrix inversion as $^{-1_m}$. The operators designate the type of their result, and hence the $*_v$ supports scalar, vector, and matrix operands, and $*_m$ supports scalar and matrix operands. Operator types facilitate the learning of the program embedding, avoiding the need to learn type propagation.

**Examples**  Programs of the form $A(BC^tD)E^{-1}$, $\vec{a}+b^{-1}\vec{c}-0\vec{e}$, $(a+b)+(c(d/e))$, $(aA+bB)C^t$ etc. can be parsed trivially to our representation, one simply needs to be able to provide a unique name for each operand and operator type (possibly via some analysis, or simple language design principles), that is avoiding to overload the semantics of operators and operands. Note the semantics is never explicitly provided to our DNN approach, it is learned by examples. There will be no example of the form e.g. $a + A$, an invalid program in our language.

We believe a sensible approach is to develop a clean, regular grammar for the language to be handled, as implicitly these are concepts the DNN will need to learn. We did so, using a classical LL(1) grammar description of our linear algebra language. This is not a requirement of our approach, as one can arrive to the desired input program graph by any means necessary, but we believe making the reasoning on the language structure simple enough is an important design aspect.

### B.2   AXIOMS OF EQUIVALENCE

A central aspect of our approach is to view the problem of program equivalence as finding a sequence of locally-correct rewrite rules that each preserve the semantics, *thereby making incremental reasoning possible*. We explicitly do not consider non-semantics-preserving "axioms". A rich structure of alternate but equivalent ways to rewrite one program to another makes the problem easier to sample and more amenable to machine learning. Semantics-preserving axioms enable incremental per-axiom reasoning, and enforce semantics preservation without overly complicated semantics analysis; while still manipulating a very rich space of transformations. To illustrate this we specifically design axioms that perform complex graph modifications, such as node deletion or creation, subtree manipulation, multi-node graph changes, etc.

A graph pattern can be viewed as a pattern-matching rule on graphs and its precise applicability criteria. It can also be viewed as a sentential form of the language grammar, e.g. `ScalarVal PlusOp ScalarVal` is a pattern, if the grammar is well formed.

**Definition 4** (Graph pattern). *A graph pattern $P$ is an unambiguous structural description of a (sub-)graph $G_P$, which can be deterministically matched in any program graph $G$. We have $P = \langle G_P, M_n, M_e \rangle$ where for each node $n_i \in N^{G_P}$, $\{n_{match}\} = M_n(n_i)$ returns the set of node*

values $n_{match}$ accepted to match $n_i$ on a graph $G$. For $n_i, n_j \in N^{G_P}$, $e_i = M_e(n_i, n_j)$ returns the set of edges between $M(n_i)$ and $M(n_j)$ to be matched in $G$. A pattern $G_P$ is matched in $G$ if (a) $\forall n_i \in G_p, \exists\, n_m = M(n_i) \in N^G$; (b) $\forall e_i \in E^{G_P}, \exists\, e_{M_n(n_i), M_n(n_j)} = M_e(n_i, n_j) \in E^G$; and (c) $\nexists e_{M_n(n_i), M_n(n_j)} \in E^G \neq M_e(n_i, n_j)$.

Note when a graph pattern models a rewrite, $M_n$ and $M_e$ are adjusted accordingly to output the rewrite of a node $n \in N^G$ into its desired value, instead of the set of acceptable nodes from $n \in N^{G_P}$.

**Definition 5** (Axiom of equivalence). *An axiom $A$ is a semantics-preserving rewrite rule $G' = A(n, G)$ that can arbitrarily modify a program graph $G$, and produces another program graph $G'$ respecting Def. 3 with identical semantics to $G$. We note $A : \langle P_{match}, P_{replace}\rangle$ an axiom, where $P_{match}, P_{replace}$ are graph patterns as per Def. 4. The application of axiom $A$ to node $n$ in $G$ is written $A(n, G)$.*

We can compose axioms to form a complex rewrite sequence.

**Definition 6** (Semantics-preserving axiom composition). *Given a sequence $S : A_1(n_1, A_2(n_2, ..., A_m(n_m, G)))$ of $m$ axioms applications. It is a semantics-preserving composition if for each $G_j = A_i(n_i, G_i) \in S$, $P_{match}^{A_i}$ succeeds on the subgraph with root $n_i$ in $G_i$, and $G_j$ is obtained by applying $P_{replace}^{A_i}$ to $n_i$.*

**Theorem 1** (Equivalence between program graphs). *Given a program $G$. If $G' = S(G)$ such that $S$ is a semantics-preserving sequence as per Def. 6, then $G \equiv G'$, they are equivalent under the axiom system used in $S$.*

This is a direct consequence of using only semantics-preserving axioms, each rewrite cannot individually alter the semantics, so such incremental composition does not. It leads to the formal problem we are addressing:

**Corollary 1** (Sufficient condition for program equivalence). *Given two programs $G, G'$. If there exist a semantics-preserving sequence $S$ such that $G' = S(G)$, then $G \equiv G'$.*

Note here $=$ means complete structural equivalence between the two graphs: they are identical in structure *and* label/node values. Determining $G = G'$ amounts to visiting both graphs simultaneously e.g. in depth-first search from the root to ensure structural equivalence, and also verifying the same node labels appear in both at the same time. This is trivally implemented in linear time in the graph size.

**Language of linear algebra expressions** We have implemented a total of 143 different axioms for our language, which are grouped for learning by our network into 14 multi-typed rewrite rules described later in Table 1. They all follow established linear algebra properties. Note different data types have different axioms following typical linear algebra rules, e.g., matrix-multiplication does not commute, but scalar and vector multiplications do. Examples of axioms include $x(yz) \to (xy)z$, $X - X \to O$, $-(\vec{x} - \vec{y}) \to \vec{y} - \vec{x}$, or $X^{t^t} \to X$, an exhaustive list is displayed in Sec. D.

In our experiments, we presume matrix and vector dimensions are appropriate for the given operation. Such dimension compatibility checks are simple to implement by e.g. introducing additional nodes in the program representation, but are not considered in our test language.

**Examples** We illustrate axiom-based rewrites using axioms presented in main Table 1. Note axiom names follow the structural changes applied. For example, we have $a + b \equiv b + a : \{a + b\} = Commute(\{+\}, \{b + a\})$. $a + b + c \equiv b + c + a : \{a + b + c\} = Commute(\{+_1\}, Commute(\{+_2\}, \{b + c + a\}))$. Note we refer to different nodes with the same symbol (e.g., $+_2$) subscripting them by their order in a DFS traversal of the program graph, starting from the unique root. We have $0 \equiv a - a : \{0\} = Cancel(\{-\}, \{a - a\})$. These can be combined in complex paths, e.g., $b + c \equiv c + b + (a - a) : \{b + c\} = Commute(\{+\}, Noop(\{+\}, Cancel(\{-\}, \{c + b + (a - a)\})))$. Such axioms are developed for scalars, matrices and vectors, and include complex rewrites such as distributivity rules and transpositions. A total of 143 axioms are used in our system.

### B.3 SPACE OF EQUIVALENCES

We now define the search space being explored in this work, i.e., the exact space of solutions on which the DNN system formally operates, and that we sample for training.

**Definition 7** (Graph of Program Equivalences). *Given a language $\mathcal{L}$. The directed graph of equivalences between programs is $GPE = \langle N^{equiv}, E^{equiv} \rangle$ such that $\forall l \in \mathcal{L}, n_l \in N^{equiv}$, and $e_{n_i,n_j}^{A_i,x} : n_i \to n_j \in E^{equiv}$ iff $n_j \equiv A_i(x, n_i)$, $\forall A_i$ in the axiom system and $x$ a position in $n_i$ where $A_i$ is applicable.*

In other words, the graph has one node per possible program in the language $\mathcal{L}$, and a single axiom application leads to connecting two nodes. We immediately note that $GPE$ is a (possibly infinite) multigraph, and contains circuits.

**Theorem 2** (Program equivalence with pathfinding). *Given two programs $n_i, n_j \in N^{equiv}$. If there is any path from $n_i$ to $n_j$ in GPE, then $n_i \equiv n_j$.*

The proof is a direct consequence of Def. 7. In this work, we randomly sample this exact graph $GPE$ to learn how to build paths between arbitrary programs, themselves represented as nodes in the $GPE$. As it is a multigraph, there will be possibly many different sequences modeled to prove the equivalence between two programs. It is sufficient to expose one to prove equivalence.

**Corollary 2** (Semantics-preserving rewrite sequence). *Any directed path in $GPE$ is a semantics-preserving rewrite sequence between the programs, described by the sequence of axioms and program position labeling the edges in this path. This sequence forms the proof of equivalence.*

We believe that ensuring there are possibly (usually) many ways to compute a proof of equivalence in our specific framework is key to enable the DNN approach to learn automatically the pathfinding algorithm for building such proofs. Other more compact representations of this space of equivalences are clearly possible, including by folding nodes in the equivalence graph for structurally-similar programs and folding equivalent paths between nodes. When building e.g. a deterministic algorithm for pathfinding, such space size reduction would bring complexity benefits Kaplan (1969); Barthou et al. (2002). We believe that for the efficient deployment of graph-to-sequence systems, exposing significant redundancy in the space facilitates the learning process. We also alleviate the need to reason on the properties of this space to find an efficient traversal heuristic.

## C   DATASET GENERATION

### C.1   GENERATION OF EXAMPLES

Machine learning benefits from large training sets, so in order to produce this data, we created algorithms that would generate programs meeting a given language grammar along with target programs which could be reached by applying a given axiom set. By creating this process, we could create as large and varied a dataset as our machine learning approach required.

Algorithm 1 provides an overview of the full program generation algorithm. For this generation process, we define a set of operations and operands on scalars, matrices, and vectors. For our process, we presume matrix and vector dimensions are appropriate for the given operation as such dimension checks are simple to implement and are not considered in our procedure. Note the token syntax here is *exactly* the one used by our system, and is *strictly* semantically equivalent to the mathematical notations used to describe these operations, e.g. $1_{\mathbb{N}}$ is 1.

- Scalar operations: `+s -s *s /s is ns`, where `is` the unary reciprical and `ns` is the unary negation.
- Matrix operations: `+m -m *m im nm tm`, where `im` is matrix inversion, `nm` negates the matrix, and `tm` is matrix transpose.
- Vector operations: `+v -v *s nv`, where `nv` is the unary negation.
- Scalars: `a b c d e 0 1`
- Matrices: `A B C D E O I`, where `O` is the empty matrix and `I` is the identity matrix.
- Vectors: `v w x y z o`, where `o` is the empty vector.
- Summary: 16 operations, 20 terminal symbols

Initially, `GenP1` is called with `GenP1("+s -s *s /s +s -s *s /s +s -s *s /s is ns +m -m *m +m -m *m +m -m *m im nm tm +v -v *v +v -v *v +v -v *v nv",0.94)"` In this initial call binary operations are repeated so that they are more likely to be created than unary operations, and the initial probability that a child of the created graph node will itself be an operation (as opposed to a terminal symbol) is set to 94%. Since the algorithm subtracts a 19% probability for children at each level of the graph, trees are limited to 7 levels.

Algorithm 1 starts execution by randomly selecting an operation from the set provided as input. When `GenP1` is called recursively, the operation set is limited such that the operation produces the correct type as output (scalar, matrix, or vector). Lines 3 through 15 of the algorithm show an example case where the `*s` operation is processed. This operation requires scalar operands. If the probability of children at this level is met, then `GenP1` is called recursively with only scalar operands available, otherwise a random scalar operand is chosen.

The text for algorithm 1 does not show the process for all operations. Certain operations, such as `*v`, have a variety of operand types that can be chosen. The `*v` operand is a multiplication which produces a vector. As such, $Av$ (matrix times vector), $bv$ (scalar times vector), or $vc$ (vector times scalar) are all valid options and will be chosen randomly.

---

**Algorithm 1:** GenP1

**Result:** Prefix notation of computation with parenthesis
**Input** : Ops, P
**Output :** (op L R) or (op L)

1   op = select randomly from Ops
2   // Create subtree for chosen op
3   **if** *op == "*s"* **then**
4     **if** *random < P* **then**
5       L = GenP1("+s -s *s /s +s -s *s /s is ns",P-0.19)
6     **else**
7       L = select random scalar operand
8     **end**
9     **if** *random < P* **then**
10      R = GenP1("+s -s *s /s +s -s *s /s is ns",P-0.19)
11     **else**
12      R = select random scalar operand
13     **end**
14     return (op L R)
15 **end**
16 // Other ops may have more complex options for children types.
17 // (For example, "*m" may have a matrix multiplied by a scalar or matrix)
18 ...

---

After generating a program which follows the grammar rules of our language, algorithm 2 will produce a new program along with a set of rewrite rules which transform the source program to the target program.

Algorithm 2 receives as input the source program (or subprogram) along with the `path` to the current root node of the source program. If the source program is a terminal symbol, the algorithm returns with no action taken. Otherwise, the program starts with an operation and the algorithm proceeds to process options for transforming the given operation. For our wholeproof10 and wholeproof5 datasets, algorithm 2 is only called once, simplifying the possible node order and proof complexity. for the axiomstep10 and axiomstep5 datasets, algorithm 2 is called multiple times, allowing for the possibility that after a path is chosen for one axiom any node can be accessed for the next axiom (including the same node).

As shown on line 10 of the algorithm, when the operation and children meet the conditions necessary for a rewrite rule (in this case `NeutralOp`), the rule is applied with some probability (in this case 50%). Note that before processing a node, the left and right operands are further analyzed to determine their operators and operands as well (or $\perp$ if the child is a terminal). Processing the left and

right operands allows for complex axioms to be applied, such as distribution or factorization. When a rule is applied, the rewrite rule is added to the rewrite rule sequence and a new target program is generated for any remaining subtrees. When creating the rewrite rules for subtrees, the `path` variable is updated as rewrites are done. In the case of `NeutralOp`, the current node is being updated, so the path is not changed. But in the case of the Commute rule, the return would be generated with `(op GenP2(R,path."left ") GenP2(L,path."right "))` which creates rewrite rules for the prior right and left operands of the `op` and updates the path used to the new node positions. In order to analyze nearly equal programs, illegal rewrites can be optionally enabled; for example, commuting a subtraction operation or mutating one operation into another. In that case, the `GenP2` process continues to create a target program, but `transform_sequence` is set to `Not_equal`.

---

**Algorithm 2:** GenP2

---

**Result:** Second program and transform_sequence
**Input** : P1, path
**Output**: P2

1 **if** *terminal symbol* **then**
2    | return P1
3 **end**
4 op = find operator of P1
5 L = find left operand of P1
6 R = find right operand of P1
7 Lop,LL,LR = operator and operands of left child
8 Rop,RL,RR = operator and operands of right child
9 // Randomly apply transform if allowed
10 **if** *random < 0.5 and ((op == "+v" and (L == "o" or R == "o")) or (op == "-v" and R == "o"))*
   **then**
11    | append path."NeutralOp " to transform_sequence
12    | // Eliminate unnecessary operator and 0 vector
13    | **if** *L == "o"* **then**
14    |   | return GenP2(R,path)
15    | **else**
16    |   | return GenP2(L,path)
17    | **end**
18 **end**

---

After these generation algorithms are run, a final data preparation process is done which prunes the data set for the learning algorithm. The pruning used on our final data set insures that the $(P1, P2)$ program pair total to 100 tokens or fewer (where a token is an operation or terminal), that the graph is such that every node is reachable from the root with a path of length 6 or less, and that there are 10 or fewer rewrite rules applied. But within these restrictions, we assert that our random production rule procedure has a non-zero probability of producing any program allowed by the grammar. Also, the pruning insures that there are no lexically equivalent programs in the process and removes some of the cases with fewer than 10 rewrite rules generated to bias the dataset to longer rewrite sequences. Table 1 details the distribution of rewrite rules created by the full process. Section D details all axioms when variable types and operators are considered.

We produce equivalent program samples by pseudo-randomly applying axioms on one randomly generated program to produce a rewrite sequence and the associated equivalent program. Given a randomly selected node in the program graph, our process checks which axiom(s) can be applied. E.g., the $+_m$ operator can have the Commute axiom applied, or depending on subtrees it may be allowed to have the Factorleft axiom applied, as discussed in Sec. 6. Generally we choose to apply or not an operator with 50% probability, so that `pe-graph2axiom` is forced to rely on analysis of the two programs to determine whether an operator is applied instead of learning a bias due to the local node features.

## C.2 Intermediate program generation

The intermediate program generation algorithm is very similar to algorithm 2. For program generation of the target program, algorithm 2 will check that a node can legally apply a given rule, apply the rule with some probability, record the action, and process the remaining program. For intermediate program generation, we begin with a $P1$ and a rewrite rule. We follow the path provided to identify the node, check that a node can legally accept a rule, apply the rule, and return the adjusted program. If a rule cannot legally be applied, $P1$ is not successfully transformed. If a rule can be legally applied to $P1$, the program is compared lexically to $P2$ and if they match then equivalence has been proven.

## C.3 Complexity of Proving Equivalence

Table 6 shows the complexity of the solution space for our problem for proofs from our AxiomStep10 test dataset up to length 7 (deterministically computing all possible programs requires too many resources for longer proof lengths). The 'All possible nodes and axioms' row includes the total number of proofs of a given length available to our problem space. The entry 5933 for a single axiom represents that for an AST depth of 7 we have 43 axioms which can be applied to all 63 possible operator nodes and 104 axioms which can be applied to the 31 nodes which possibly have child operator nodes themselves: 63*43+31*104=5933. Subsequent columns can select repeatedly from the same set growing as $5933^2$ to $5933^7$. The 'sample node + axiom group' row is based on our 10,000 sample test dataset and represents the possible selection of any of the 14 axiom groups being applied to any node in the program. The 'sample node + legal axiom' row represents only legal node plus legal axiom group being applied and effectively represents the total number of programs derivable from the start program in the test dataset. The final row 'Sample derivable unique programs' represents the total number of programs derived from legal node and axiom sequences which are lexically unique.

Table 6: Counts for equivalence proof possibilities

| Proof description | Proof length in axioms | | | | | | |
|---|---|---|---|---|---|---|---|
| | 1 | 2 | 3 | 4 | 5 | 6 | 7 |
| All Possible nodes and axioms | 5933 | 3.5E+07 | 2.1E+11 | 1.2E+15 | 7.4E+18 | 4.4E+22 | 2.6E+26 |
| Sample Node + Any Axiom | 226 | 46900 | 1.5E+07 | 8.8E+09 | 5.0E+12 | 3.3E+15 | 2.7E+18 |
| Sample Node + Legal Axiom | 11.2 | 77.8 | 931 | 15812 | 3.4E+05 | 8.2E+06 | 1.8E+08 |
| Unique Programs from Sample | 9.2 | 47.4 | 264 | 1574 | 10052 | 65176 | 4.6E+05 |

# D Language and Axioms for Complex Linear Algebra Expressions

We now provide the complete description of the input language for multi-type linear algebra expressions we use to evaluate our work, and the complete list of all axioms that are used to compute equivalence between programs.

**Variable types** We model programs made of scalars, vectors and matrices. We limit programs to contain no more than 5 distinct variable names of each type in a program:

- Scalar variables are noted $a$, $b$, ..., $e$.
- Vector variables are noted $\vec{v}$, $\vec{w}$, ..., $\vec{z}$.
- Matrix variables are noted $A$, $B$, ..., $E$.

Note we also explicitly distinguish the neutral and absorbing elements for scalars and matrices, e.g. $1 = 1_{\mathbb{N}}$. This enables the creation of simplification of expressions as a program equivalence problem, e.g. if $A + B - (B + A) = 0_{\mathbb{K} \times \mathbb{K}}$

**Unary operators** We model 6 distinct unary operators, all applicable to any variable of the appropriate type:

- `is(a)` $= a^{-1}$ is the unary reciprocal for scalars, `im(A)` $= A^{-1}$ is matrix inverse.
- `ns(a)` $= -a$ is unary negation for scalars, `nv(v)` $= -\vec{v}$ for vectors, `nm(M)` $= -M$ for matrices.
- `tm(M)` $= M^t$ is matrix transposition.

**Binary operators**  We model 10 distinct binary operators that operate on two values. 7 operators require the same type for both operands, while 3 enable multi-type operands (e.g., scaling a matrix by a scalar). Note we do not consider potential vector/matrix size compatibility criterion for these operators, in fact we do not represent vector or matrix sizes at all in our language, for simplicity.

- $+\text{s}(\text{a, b}) = a + b$, the addition on scalars, along with $-\text{s}(\text{a,b}) = a - b$, $\star\text{s}(\text{a,b}) = a * b$ and $/\text{s}(\text{a,b}) = a/b$.
- $+\text{v}(\text{ v, w}) = \vec{v} + \vec{w}$, the addition on vectors, along with $-\text{v}(\text{ v , w}) = \vec{v} - \vec{w}$, $\star\text{v}(\text{ v, w}) = \vec{v}.\vec{w}$ the dot product between two vectors, producing a scalar.
- $+\text{m}(\text{A, B}) = A + B$, the addition on matrices, along with $-\text{m}(\text{A, B}) = A - B$, and $\star\text{m}(\text{A, B}) = AB$ the product of matrices.
- $\star\text{m}(\text{a,A}) = a\dot{A}$ and $\star\text{m}(\text{A,a}) = A\dot{a}$ are used to represent scaling a matrix by a scalar.
- $\star\text{m}(\text{v,A}) = \vec{v}A$ represents a vector-matrix product.
- $\star\text{v}(\text{a,v}) = a\vec{v}$ and $\star\text{v}(\text{v,a}) = \vec{v}a$ represent scaling a vector by a scalar.

**List of axioms of equivalence**  Tables 7-8 show the full 147 axioms supported by our rewrite rules. Many rewrite rules can be applied to all 3 variable types as well as multiple operator types.

| Rewrite Rule | ID | Example(s) | Rewrite Rule | ID | Example(s) |
|---|---|---|---|---|---|
| Cancel | 1 | (a - a) → 0 | AbsorbOp | 28 | (a * 0) → 0 |
| | 2 | (b/b) → 1 | | 29 | (0 * a) → 0 |
| | 3 | (A - A) → O | | 30 | (A * 0) → O |
| | 4 | $(A * A^{-1}) \to I$ | | 31 | (0 * A) → O |
| | 5 | $(A^{-1} * A) \to I$ | | 32 | (A * O) → O |
| | 6 | (v - v) → o | | 33 | (O * A) → O |
| NeutralOp | 7 | (a + 0) → a | | 34 | (A * o) → o |
| | 8 | (0 + a) → a | | 35 | (a * o) → o |
| | 9 | (a - 0) → a | | 36 | (o * a) → o |
| | 10 | (a * 1) → a | | 37 | (0 * v) → o |
| | 11 | (1 * a) → a | | 38 | (v * 0) → o |
| | 13 | (a / 1) → a | | 39 | (O * v) → o |
| | 14 | (A + O) → A | Commute | 40 | (a + b) → (b + a) |
| | 15 | (O + A) → A | | 41 | (a * b) → (b * a) |
| | 16 | (A - O) → A | | 42 | (A + B) → (B + A) |
| | 17 | (A * I) → A | | 43 | (A * a) → (a * A) |
| | 18 | (I * A) → A | | 44 | (a * A) → (A * A) |
| | 19 | (v + o) → v | | 45 | (A * O) → (O * A) |
| | 20 | (o + v) → v | | 46 | (O * A) → (A * O) |
| | 21 | (v - o) → v | | 47 | (A * I) → (I * A) |
| DoubleOp | 22 | -(-a)) → a | | 48 | (I * A) → (A * I) |
| | 23 | $(a^{-1})^{-1} \to a$ | | 49 | (v + w) → (w + v) |
| | 24 | $-(-A) \to A$ | | 50 | (v * a) → (a * v) |
| | 25 | $(A^{-1})^{-1} \to A$ | | 51 | (a * v) → (v * a) |
| | 26 | $(A^t)^t \to A$ | DistributeLeft | 52 | (a + b)c → ac + bc |
| | 27 | $-(-v)) \to v$ | | 53 | (a - b)c → ac - bc |
| DistributeRight | 64 | a(b + c) → ab + ac | | 54 | (a + b)/c → a/c + b/c |
| | 65 | a(b - c) → ab - ac | | 55 | (a - b)/c → a/c - b/c |
| | 66 | a(v + w) → av + av | | 56 | (v + w)*a → va + wa |
| | 67 | a(v - w) → av - av | | 57 | (v - w)*a → va - wa |
| | 68 | A(B + C) → AB + AC | | 58 | (A + B)C → AC + BC |
| | 69 | A(B - C) → AB - AC | | 59 | (A - B)C → AC - BC |
| | 70 | a(B + C) → aB + aC | | 60 | (A + B)v → Av + Bv |
| | 71 | a(B - C) → aB - aC | | 61 | (A - B)v → Av - Bv |
| | | | | 62 | (A + B)a → Aa + Ba |
| | | | | 63 | (A - B)a → Aa - Ba |

Table 7: Full axiom count when all type options and other supported permutations are included (part 1 of 2)

| Rewrite Rule | ID | Example(s) | Rewrite Rule | ID | Example(s) |
|---|---|---|---|---|---|
| FactorLeft | 72 | ab + ac → a(b+c) | AssociativeRight | 113 | (a+b)+c → a+(b+c) |
| | 73 | ab - ac → a(b-c) | | 114 | (a+b)-c → a+(b-c) |
| | 74 | AB + AC → A(B+C) | | 115 | (ab)c → a(bc) |
| | 75 | AB - AC → A(B-C) | | 116 | (A+B)+C → A+(B+C) |
| | 76 | Av + Aw → A(v+w) | | 117 | (A+B)-C → A+(B-C) |
| | 77 | Av - Aw → A(v-w) | | 118 | (AB)C → A(BC) |
| | 78 | Aa + Ab → A(a+b) | | 119 | (AB)a → A(Ba) |
| | 79 | Aa - Ab → A(a-b) | | 120 | (Aa)B → A(aB) |
| | 80 | va + vb → v(a+b) | | 121 | (aA)B → a(AB) |
| | 81 | va - vb → v(a-b) | | 122 | (Av)a → A(va) |
| FactorRight | 82 | ac + bc → (a+b)c | | 123 | (Aa)v → A(av) |
| | 83 | ac - bc → (a-b)c | | 124 | (aA)v → a(Av) |
| | 84 | a/c + b/c → (a+b)/c | | 125 | (va)b → v(ab) |
| | 85 | a/c - b/c → (a-b)/c | | 126 | (av)b → a(vb) |
| | 86 | AC + BC → (A+B)C | | 127 | (ab)v → a(bv) |
| | 87 | AC - BC → (A-B)C | | 128 | (v+w)+x → v+(w+x) |
| | 88 | Av + Bv → (A+B)v | | 129 | (v+w)-x → v+(w-x) |
| | 89 | Av - Bv → (A-B)v | FlipLeft | 130 | -(a - b) → b-a |
| | 90 | Aa + Ba → (A+B)a | | 131 | $(a/b)^{-1}$ → b/a |
| | 91 | Aa - Ba → (A-B)a | | 132 | $-(A - B)$ → (B - A) |
| | 92 | va + wa → (v+w)a | | 133 | $-(v - w)$ → (w - v) |
| | 93 | va - wa → (v-w)a | FlipRight | 134 | a/(b/c) → a(c/b) |
| AssociativeLeft | 94 | a+(b+c) → (a+b)+c | | 135 | $a/(b^{-1})$ → ab |
| | 95 | a+(b-c) → (a+b)-c | | 136 | a-(b-c) → a+(c-b) |
| | 96 | a(bc) → (ab)c | | 137 | a-(-b) → a+b |
| | 97 | a(b/c) → (ab)/c | | 138 | A-(B-C) → A+(C-B) |
| | 98 | A+(B+C) → (A+B)+C | | 139 | A-(-B) → A+B |
| | 99 | A+(B-C) → (A+B)-C | | 140 | v-(w-x) → v+(x-w) |
| | 100 | A(BC) → (AB)C | | 141 | v-(-w) → v+w |
| | 101 | A(Ba) → (AB)a | Transpose | 142 | $(AB) \rightarrow (B^t A^t)^t$ |
| | 102 | A(aB) → (Aa)B | | 143 | $(A + B) \rightarrow (A^t + B^t)^t$ |
| | 103 | a(AB) → (aA)B | | 144 | $(A - B) \rightarrow (A^t - B^t)^t$ |
| | 104 | A(Bv) → (AB)v | | 145 | $(AB)^t \rightarrow B^t A^t$ |
| | 105 | A(va) → (Av)a | | 146 | $(A + B)^t \rightarrow A^t + B^t$ |
| | 106 | A(av) → (Aa)v | | 147 | $(A - B)^t \rightarrow A^t - B^t$ |
| | 107 | a(Av) → (aA)v | | | |
| | 108 | v+(w+x) → (v+w)+x | | | |
| | 109 | v+(w-x) → (v+w)-x | | | |
| | 110 | v(ab) → (va)b | | | |
| | 111 | a(vb) → (av)b | | | |
| | 112 | a(bv) → (ab)v | | | |

Table 8: Full axiom count when all type options and other supported permutations are included (part 2 of 2)

# E    DETAILS ON NEURAL NETWORK MODEL

Figure 4 overviews the entire `pe-graph2axiom` architecture including sample generation, the graph-to-sequence network, the intermediate program generation, and lexical equivalence checker. In this section we will discuss the implementation details of these components.

**Graph neural network internal representation**    The sample generation discussed in section 4 provides input to the Node Initialization module in Fig. 4 to create the initial state of our graph neural network. For each node in the program graph, a node will be initialized in our graph neural network. Each node has a hidden state represented by a vector of 256 floating point values which are used to create an embedding for the full meaning of the given node. Initially all 256 dimensions of the hidden states of the nodes are set to zero except for 2. Given $N$ tokens in our input program language, one of the dimensions from 1 through $N$ of a node will be set based on the token at the program position that the node represents. For example, if the scalar variable $a$ is assigned to be token 3 in our language, then the $a$ nodes of Fig. 3 would have their 3rd dimension initialized to 1.0. This is a

one-hot encoding similar to that used in neural machine translation models which leverage Word2vec Mikolov et al. (2013). The second non-zero dimension in our node initialization indicates the tree depth, with the root for the program being at depth 1. We set the dimension $N+depth$ to 1.0; hence, the $a$ nodes in Fig 3, which vary from level 2 or 3 in the graph, would set dimension $N + 2$ or $N + 3$ to 1. In addition to nodes correlating to all tokens in both input programs, we initialize a root node for program comparison which has edges connecting to the root nodes of both programs. The root node does not represent a token from the language, but it is initialized with a 1.0 in a hidden state dimension reserved for its identification.

For a graph neural network, the edge connections between nodes are a crucial part of the setup. In particular, to match the formulation of our problem, we must ease the ability of the network to walk the input program graphs. We therefore designed a unified graph input, where both program graphs are unified in a single graph using a single connecting root node; and where additional edges are inserted to make the graph fully walkable.

In our full model, we support 9 edge types and their reverse edges. The edge types are: 1) left child of binary op, 2) right child of binary op, 3) child of unary op, 4) root node to program 1, 5) root node to program 2, 6-9) there are 4 edge types for the four node grandchilden (LL, LR, RL, RR). After the node hidden states and edge adjacency matrix are initialized, the network is ready to begin processing. This initial state is indicated in figure 5 by the solid circles in the lower left of the diagram.

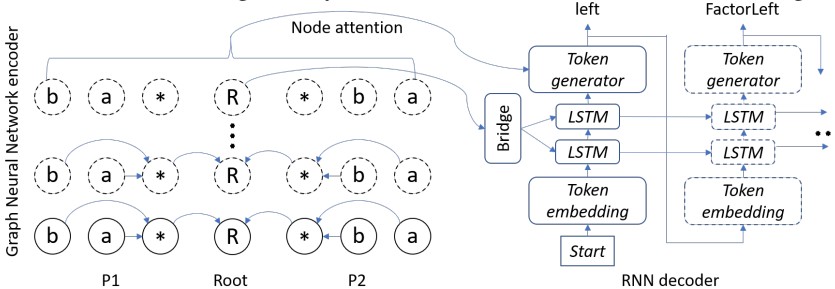

Figure 5: Graph-to-sequence neural network data flow details.

**Beam search**    A typical approach when using sequence-to-sequence systems is to enable *beam search*, the process of asking for multiple answers to the same question to the network. It is particularly relevant when creating outputs which can be automatically checked Chen et al. (2019); Ahmed et al. (2018). Beam search can be viewed as proposing multiple possible axioms to apply. Given the stochastic nature of generation model, a beam width of $n$ can be thought of as creating the $n$ most likely sequences given the training data the model as learned on. Each proposal can be checked for validity, the first valid one is outputted by the system, demonstrating equivalence. Our system builds on the neural network beam search provided by OpenNMT to create a 'system beam search' of variable width. In particular, we set the OpenNMT network beam search to 3, which constrains the token generator to produce 3 possible axiom/node proposals for a given pair of input programs. Using these 3 proposals, when our system beam width is 10, we build up to 10 intermediate programs that are being processed in the search for a proof. To illustrate with a system beam width of 5, after $P1$ and $P2$ are provided to the neural network, 3 possible intermediate programs may be created (so long as all axioms are legal and don't produce duplicates). After those 3 intermediates are processed, 9 possible new intermediates are created, all of which are checked for lexical equivalence with $P2$, but only 5 of which are fed back into the neural network for further axiom generation. This process is continued for up to 12 axioms at which point the system concludes an equivalence proof cannot be found and the programs are likely not equivalent. We evaluate in Sec. 6 beam sizes ranging from 1 to 10, showing higher success with larger beams.

## F    DETAILS ON EXPERIMENTAL RESULTS

### F.1    COMPLEMENTARY RESULTS AND OBSERVATIONS

Table 9 describes part of our neural network hyperparameter tuning showing that our golden model has as high a result as other variations explored. Note that the validation token accuracy is not too high (it's not above 90%) despite the ability to predict full correct proofs with over 93% accuracy. This is

because the training dataset can have multiple examples of axioms given similar input programs. For example, proving "(a+b)(c+d) = (b+a)(d+c)" requires commuting the left and right subexpressions. The training dataset could have similar programs which are sometimes transformed first with a right Commute and then a left or vice-versa. Given this data, the network would learn to apply one or the other (it would not get trained to use associativity for these program pairs for example), hence the actual output given may or may not match the validation target axiom. We will discuss this further in section F.2.

Table 9: Hyperparameter experiments. Summary of best validation token accuracy result after 2 runs for up to 100,000 training iterations. The golden model has 256 graph nodes and decoder dimensions, 2 decoder LSTM layers, starts training with a learning rate of 0.8, and uses 10 steps to stabilize the GGNN encoder.

| Parameter | Value | Validation token accuracy |
|---|---|---|
| Golden model | | 83.89 |
| Graph node and decoder LSTM dimension | 192 | 83.89 |
| | 320 | 83.58 |
| Decoder LSTM layers | 1 | 83.53 |
| Initial learning rate | 0.75 | 83.76 |
| | 0.85 | 83.57 |
| GGNN stability steps | 12 | 83.19 |
| | 8 | 83.61 |

**Training convergence**   Since our model trains on axiomatic proofs which may vary in order (allowing 2 or 3 options to be correct and occur in the training set), we see our training and token accuracies plateau below 90% during training for AxiomStep10 as shown in Figure 6. Full testset proof accuracies for beam width 10 exceed 90%, but also plateau along with the training and validation results. This result differs from our WholeProof10 training, which achieves training and validation accuracies above 96% because the expected axiom sequence is more predictable, but as we have seen less generalized.

As another observation on generalization and overfitting, we note that figure 6 shows a slight separation between the training and validation accuracies starting at around iteration 180,000. While the training accuracy rises slowly, validation accuracy plateaus, indicating slight overfitting on the training data. Yet our model continues to slowly increase in quality, with the model snapshot that scores best on both validation and test accuracies occurring at iteration 300,000. This is our golden model, with 93.1% of P1 to P2 proofs accurately found using beam width 10.

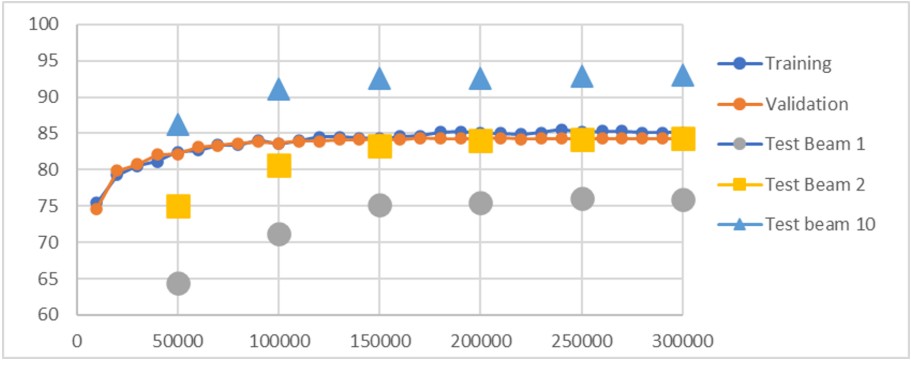

Figure 6: Model training percentage accuracy up to 300,000 iterations on AxiomStep10. Training and Validation accuracies are per-token on the target axioms in the samples. Test accuracies are for full correct proofs of P1 to P2.

**Testing simpler models**   In addition to the sequence-to-sequence and graph-to-sequence models, we explored a feed-forward equal/not equal classifier on a simple version of our language. That model uses an autoencoder on the program to find an embedding of the program and then a classifier

based on the program embeddings found. It achieves a 73% accuracy on identifying equivalent pairs in the test data, which, as expected, is much lower than the full proof rate of 93% achieved with a graph-to-sequence proof generator on our full language. This simple experiment highlights the importance of a system which prevents the false positives which a classifier might have by creating a verifiable proof.

We explore initial language generation using a simple language in order to assess feasibility of different approaches. For fine tuning network parameters and architectural features, we add more complexity to the language as shown in table 10. Language IDs 1 through 3 are all based on a simple grammar which only allows the "+" or "-" operators on scalar variables labeled a through j. The only axiom is `Commute`, which can be applied on up to 3 nodes in language IDs 2 and 3. Language ID 4 adds the scalar constants 0 and 1, scalar operations * and /, and 4 more axioms. We perform a fair amount of network development on this model in an effort to maintain high accuracy rates. Language ID also 4 expands the operands to 3 types and hence the number of operators also increases. To speed up model evaluation, we reduced the program length for IDs 5, 6, and 7, allowing us to train larger data sets for more epochs. ID 7 is a forward looking-model which makes a minor increment to the language to support the analysis of loop rolling and unrolling, discussed further in section F.3. ID 8 is the WholeProof5 model in relation to these early experiments.

| ID | Description | # Operators | # Axioms in language | # Operands | Program length | Rewrite rule axioms | Training set size | Percent matching with beam width 1 | Percent matching with beam width 10 |
|---|---|---|---|---|---|---|---|---|---|
| 1 | Rewrite sequence is 2 Commutes | 2 | 1 | 10 | 5-24 | 2 | 80K | 98.9 | 99.8 |
| 2 | Rewrite sequence is 3 Commutes | 2 | 1 | 10 | 7-45 | 3 | 80K | 91.4 | 99.0 |
| 3 | Rewrite sequence 1-3 Commutes | 2 | 1 | 10 | 3-45 | 1-3 | 180K | 97.1 | 99.2 |
| 4 | Commute, NeutralOp, Cancel, DistributeLeft, DistributeRight | 4 | 5 | 12 | 3-45 | 1-3 | 180K | 93.1 | 97.4 |
| 5 | Scalars, Vectors, and Matrices | 16 | 5 | 20 | 3-30 | 1-5 | 250K | 88.3 | 95.6 |
| 6 | 13 Axioms | 16 | 13 | 20 | 3-30 | 1-5 | 400K | 85.5 | 95.5 |
| 7 | Add loop axioms | 18 | 15 | 20 | 3-30 | 1-5 | 400K | 83.8 | 94.7 |
| 8 | 14 Axioms | 16 | 14 | 20 | 3-50 | 1-5 | 1M | 81.8 | 94.4 |

Table 10: Description and results for various language complexities studied with WholeProof models.

We designed our datasets in section 4 with the goal of using the varied models to understand the generalizability of `pe-graph2axiom` and to show that our model is not overfitting on training data. For these next experiments, all results of for beam width 10, which provides for a neural-network directed search of up to 10 axiomatic proofs of equivalence for each program pair. Recall that our most complex dataset is AxiomStep10 which includes $(P1, P2, S)$ samples requiring up to 10 rewrite rules, $P1$ and $P2$ can have up to 50 AST nodes each, and an AST depth of up to 7. AxiomStep5 has samples requiring up to 5 rewrite rules, $P1$ and $P2$ can have up to 25 AST nodes each, and an AST depth of up to 6. Tables 11 and 12 (repeated from main paper below) demonstrate the ability of a model trained on AxiomStep5 to perform well on the larger distribution of programs from AxiomStep10, implying that the model has generalized well to our program equivalence problem and that `pe-graph2axiom` does not overfit its response to merely the training set distribution.

Table 12 illustrates the ability of a model trained on AxiomStep5 (i.e., limited to proofs of length 5) to perform well when evaluated on the more complex AxiomStep10, which includes proofs of unseen length of up to 10. The robustness to the input program complexity is illustrated with the 86% pass rate on AST depth 7, for the model trained on AxiomStep5 which never saw programs of depth 7 during training.

As an indication of the breadth of equivalent programs represented by AxiomStep10 relative to WholeProof10, table 13 shows the full detail of models trained on all 4 datasets when tested on test data from all 4 datasets. AxiomStep10, while training on our broadest dataset in which axioms can

Table 11: Generalizing to longer P1 inputs. Percentage pass rates for equivalence proofs with P1 having increasing program graph nodes. The model trained with the AxiomStep5 dataset had no training examples more than 25 program graph nodes yet it performs relatively well on these more complex problems. The furthest right column shows the `pe-graph2axiom` model results on the most complex dataset.

| P1 nodes | Testset Sample Count | | Model trained on AxiomStep5 | | Model trained on AxiomStep10 | |
|---|---|---|---|---|---|---|
| | AS5 | AS10 | AS5 | AS10 | AS5 | AS10 |
| 1-5 | 231 | 109 | 100 | 100 | 100 | **100** |
| 6-10 | 2147 | 1050 | 100 | 99 | 99 | **99** |
| 11-15 | 3980 | 2175 | 99 | 96 | 99 | **96** |
| 16-20 | 2583 | 2327 | 98 | 92 | 98 | **93** |
| 21-25 | 1059 | 1989 | 97 | 89 | 98 | **92** |
| 26-30 | 0 | 1229 | N/A | 83 | N/A | **90** |
| 31-35 | 0 | 698 | N/A | 78 | N/A | **88** |
| 36-40 | 0 | 304 | N/A | 74 | N/A | **87** |
| 41-45 | 0 | 101 | N/A | 68 | N/A | **84** |
| 46-50 | 0 | 27 | N/A | 67 | N/A | **85** |
| All | 10000 | 10000 | 99 | 90 | 99 | **93** |

Table 12: Performance vs. AST size: counts and percentage pass rates.

| AST depth | Testset Sample Count | | Model trained on AxiomStep5 | | Model trained on AxiomStep10 | |
|---|---|---|---|---|---|---|
| | AS5 | AS10 | AS5 | AS10 | AS5 | AS10 |
| 2 | 5 | 3 | 100 | 100 | 100 | 100 |
| 3 | 306 | 133 | 100 | 100 | 100 | 100 |
| 4 | 1489 | 577 | 100 | 99 | 99 | 99 |
| 5 | 4744 | 1844 | 99 | 94 | 98 | 95 |
| 6 | 3456 | 4308 | 98 | 90 | 98 | 93 |
| 7 | 0 | 3135 | n/a | 86 | n/a | 92 |
| All | 10000 | 10000 | 99 | 90 | 99 | **93** |

be applied to nodes repeatedly and in variable order, achieves a 93% average success rate. 72% of the proofs of length 6 from the WholeProof10 testset were solved by the model trained on WholeProof10, but only 5% of such proofs from AxiomStep10 were, suggesting the method of generating AxiomStep pairs covers the problem space more thoroughly.

The complete result for the WholeProof10 model on the WholeProof10 dataset was 8,388 out of 10,000 program pairs had a correct proof found; of those, 8,350 were the exact proof created during $P1, P2$ generation, implying that WholeProof10, while performing well on its own testset distribution, is not learning to generalize to alternative proof paths.

**Manual verifications** We conducted a series of manual verifications of the system used to produce all the above results. First, we are happy to confirm that most likely $AB \neq BA$ given no verifiable equivalence sequence was produced, but that provably $ab = ba$ indeed. We also verified that $A^{t^t}(B + C - C) = AB$, and that $AB\vec{v} - AB\vec{w} = AB(\vec{v} - \vec{w})$ which would be a much faster implementation. The system correctly suggests that $AB\vec{v} - BA\vec{w} \neq AB(\vec{v} - \vec{w})$. We ensured that $A^t(AA^t)^{-1}A \neq A^t(AA^{-1})^t A$, from a typo we once made when typing the computation of an orthonormal sub-space. We also verified that indeed $AB + AC + aD - aD = A(B + C)$.

**Generalizing variable types** We explored the ability of the model to understand variable typing by training a model with the AxiomStep10 distribution but with no samples that included the scalar variable 'e' and scalar multiplication $*_s$. This removed about 50% of the training set, as longer programs were often included both tokens. When tested with the unaltered AxiomStep10 test set and beam width 10, test samples that included a scalar variable not 'e' and $*_s$ were proven equal

Table 13: Generalizing to longer proofs. Percentage pass rates for equivalence proofs of increasing axiom counts when testing each of 4 datasets on models trained using each of 4 datasets.

| Axiom Count in Proof | Model trained on WholeProof5 (WP5) | | | | Model trained on WholeProof10 (WP10) | | | | Model trained on AxiomStep5 (AS5) | | | | Model trained on AxiomStep10 (AS10) | | | |
|---|---|---|---|---|---|---|---|---|---|---|---|---|---|---|---|---|
| | WP5 | WP10 | AS5 | AS10 | WP5 | WP10 | AS5 | AS10 | WP5 | WP10 | AS5 | AS10 | WP5 | WP10 | AS5 | AS10 |
| 1 | 100 | 100 | 100 | 99 | 100 | 100 | 100 | 100 | 100 | 100 | 100 | 100 | 100 | 100 | 100 | **100** |
| 2 | 99 | 98 | 66 | 64 | 99 | 99 | 65 | 63 | 100 | 99 | 100 | 99 | 100 | 100 | 100 | **100** |
| 3 | 98 | 94 | 34 | 33 | 97 | 95 | 33 | 33 | 100 | 98 | 99 | 98 | 100 | 99 | 99 | **99** |
| 4 | 93 | 84 | 16 | 15 | 90 | 88 | 16 | 15 | 98 | 95 | 98 | 97 | 99 | 98 | 98 | **98** |
| 5 | 84 | 70 | 8 | 7 | 84 | 82 | 8 | 7 | 96 | 91 | 96 | 95 | 97 | 95 | 96 | **96** |
| 6 | | 14 | | 4 | | 72 | | 5 | | 81 | | 88 | | 90 | | **93** |
| 7 | | 0 | | 1 | | 63 | | 2 | | 67 | | 81 | | 83 | | **87** |
| 8 | | 0 | | 0 | | 54 | | 1 | | 54 | | 75 | | 73 | | **82** |
| 9 | | 0 | | 0 | | 47 | | 0 | | 35 | | 64 | | 63 | | **74** |
| 10 | | 0 | | 0 | | 34 | | 0 | | 24 | | 57 | | 46 | | **66** |
| All | 95 | 66 | 44 | 27 | 94 | 84 | 44 | 27 | 99 | 87 | 99 | 90 | 99 | 93 | 99 | **93** |

90% of the time; test samples that included 'e' and $*_s$ were also proven equal 90% of the time. For beam width 1 the proof success rates were 72% and 70% for without and with 'e', implying that the heavily biased training set did have a small effect on the system generalization. `pe-graph2axiom` was still able to generalize the relation of 'e' to the $*_s$ operator given that 'e' was used in contexts similar to other scalar variables in the training samples that were provided, implying it was forming an internal representation of a 'scalar' type by learning from examples.

## F.2 LEARNING THAT MULTIPLE AXIOM CHOICES ARE POSSIBLE

Our AxiomStep10 model is trained on axioms which may be applied in varying order in the training set. For example, $((a+b)*(c+d)) = ((b+a)*(d+c))$ may have the training data to Commute the left node $a+b$ first and then $c+d$ second; in the same dataset, $((a+e)*(b+c)) = ((e+a)*(c+b))$ might occur and the training data has the right node Commuted first. In this way, we expect the model to learn that either commuting the left or right node is a proper first axiom choice. Table 14 explores the ability of the model to produce such axiom proposals. Given 5 scalar variables, there are 120 possible expressions where two 2-variable additions are multiplied together such as $((a+b)*(c+d))$. We consider here all 120 program pairs in which the left and right additions are commuted. The table shows which axioms and positions are recommended by the graph-to-sequence neural network model within the `pe-graph2axiom` system as most probably moving the 2 programs closer to equivalence by the beam width 3 on this problem. Note that the 2 correct axioms are always within the top 3 choices and the other 2 axioms (Commute and DistributeLeft on the root), while not necessary for this problem, are at least legal choices for axioms within our expression language.

The results in table 14 relate to the value of our approach in relation to reinforcement learning models for proof generation Fawzi et al. (2019) Bansal et al. (2019). To make an analogy with reinforcement learning, in our training, the world 'state' is presented as a $P1, P2$ pair and the system must learn to produce an axiom at a location which performs an 'action' on the 'state' of $P1$ in a predictable way. Unlike reinforcement learning, we do not produce a reward function and our system cannot learn from a poor reward produced by an incorrect axiom. However, we have demonstrated that our system, as it is presented with a wide distribution of $(P1, P2, S)$ tuples to train on, learns a probability distribution of possibly correct axioms to produce for a given program pair. There may be value in combining our graph-neural-network within a reinforcement learning framework that used a hindsight mechanism Andrychowicz et al. (2017) to learn from every attempted axiom, but it is not immediately obvious that our approach of learning only from examples of successful equivalence proofs would be improved.

**Exploration of alternate designs**    In order to design the system, we explored parts of the design space quickly and performed several single training run comparisons between 2 options, as shown in Table 15.

Table 14: Learning multiple output options. When considering scalar expressions that can be proven equivalent by commuting the left and right subexpressions, such as $(a + b)(c + d) = (b + a)(d + c)$, `pe-graph2axiom` learns that either the left or right commute can occur first. The columns show counts for axioms and locations proposed by the token generator with beam width of 3 when given 120 different scalar expression pairs.

| Beam
position | Axiom | | | |
| | Commute
left child | Commute
right child | Commute
root | DistributeLeft
root |
| --- | --- | --- | --- | --- |
| First | 49 | 35 | 36 | 0 |
| Second | 58 | 59 | 3 | 0 |
| Third | 13 | 26 | 45 | 36 |
| Any of top 3 | 120 | 120 | 84 | 36 |

In cases where 2 options were similar, we chose the model which ran faster, or run the models a second time to get a more precise evaluation, or use our experience based on prior experiments to select an option.

Table 15: Example explorations as a single feature or parameter is changed. Each comparison is a distinct experiment, as the entire network and language used was being varied.

| Options compared | Match
beam 1 | Match
beam 10 |
| --- | --- | --- |
| 1 layer LSTM vs | 198 | 1380 |
| 2 layer LSTM vs | 5020 | 9457 |
| 3 layer LSTM | 4358 | 8728 |
| No edges to grandchild nodes vs | 9244 | 9728 |
| Edges to grandchild nodes | 9284 | 9774 |
| Encoder->Decoder only root node vs | 8616 | 9472 |
| Encoder->Decoder avg all nodes | 7828 | 9292 |

Experiments such as these informed our final network architecture. For example, in `pe-graph2axiom`, we include 4 edges with learnable weight matrices from a node to its grandchildren because such edges were found to improve results on multiple runs. Li et al. Li et al. (2019) discusses the importance of selecting the optimal process for aggregating the graph information hence we explore that issue for our network. Our approach uses the root comparison node to create aggregate the graph information for the decoder as it performs better than a node averaging.

**Including Not_equal option** Table 16 analyzes the challenge related to a model which only predicts Equal or Not_equal for program pairs along with various options which produce rewrite rules which can be checked for correctness. In all 4 output cases shown, 2 programs are provided as input. These programs use an earlier version of our language model with 16 operators, 13 core axioms, and 20 operands generated with a distribution similar to WholeProof5.

Table 16: Table showing alternate options for handling not equal programs

| Network
output
Description | Actual | Predicted
NotEq | Predicted
Rules
or Eq | Correct
Rewrite
Rules |
| --- | --- | --- | --- | --- |
| Eq or NotEq,
Beam width 1 | Eq
NotEq | 5.4%
90.4% | 94.6%
9.6% | N/A
N/A |
| Rules or NotEq,
Beam width 1 | Eq
NotEq | 6.6%
90.9% | 93.4%
9.1% | 70.7%
N/A |
| Rules only,
Beam width 1 | Eq
NotEq | N/A
N/A | 100%
N/A | 87.8%
N/A |
| Rules only,
Beam width 10 | Eq
NotEq | N/A
N/A | 100%
N/A | 96.2%
N/A |

For the first output case, the output sequence to produce is either `Equal` or `Not_equal`. Given a false positive rate of 9.6%, these results demonstrate the importance of producing a verifiable proof of equivalence when using machine learning for automated equivalence checking. For the second output case, the model can produce either `Not_equal` or a rewrite rule sequence which can be checked for correctness. The source programs for the first and second case are identical: 250,000 equivalent program pairs and 250,000 non-equivalent program pairs. In the second case, the false positive rate from the network is 9.1% (rules predicted for Not_equal programs), but the model only produces correct rewrite rules between actual equivalent programs in 70.7% of the cases.

One challenge with a model that produce rules or `Not_equal` is that beam widths beyond 1 are less usable. Consider that with a beam width of 1, if the network predicts `Not_equal` then the checker would conclude the programs are not equal (which is correct for 90.9% of the actually not equal programs). With a beam width of 10, there would be more proposed rewrite rules for equal programs to test with, but if 1 of the 10 proposals is `Not_equal`, should the checker conclude they are not equal? Or should the the checker only consider the most likely prediction (beam width 1) when checking for non-equivalence? The third and fourth network output cases provide an answer. For these 2 cases, the training set is 400,000 equivalent program pairs - none are non-equivalent. 250,000 of these pairs are identical to the equivalent programs in the first 2 cases, and 150,000 are new but were produced using the same random generation process. Note that by requiring the network to focus only on creating rewrite rules, beam width 1 is able to create correct rewrite rules for 87.8% of the equivalent programs. And now, since we've remove the confusion of the `Not_equal` prediction option, beam width 10 can be used to produce 10 possible rewrite rule sequences and in 96.2% of the cases these rules are correct. Hence, we propose the preferred use model for pe-graph2axiom is to always use the model which is trained for rule generation with beam width 10 and rely on our rule checker to prevent false positives. From the 10 rewrite rule proposals, non-equivalent programs will never have a correct rewrite rule sequence produced, hence we guarantee there are no false positives.

### F.3 AN EXAMPLE OF BACK-EDGE IN THE PROGRAM GRAPH

Figure 7 shows an example of DoX and DoHalf. The new operators result in 2 new edges in our graph representation (along with 2 new back-edges): there is a 'loopbody' edge type from the loop operator node to the start of the subgraph, and there is a 'loopfeedback' edge type from the variable which is written to each loop iteration. These 2 edge types are shown in the figure. The new $Dohalf$ axiom intuitively states that $DoX(g(y)) = DoHalf(g(g(y)))$ (where $y$ is the variable reused each iteration), and $Dox$ states the reverse.

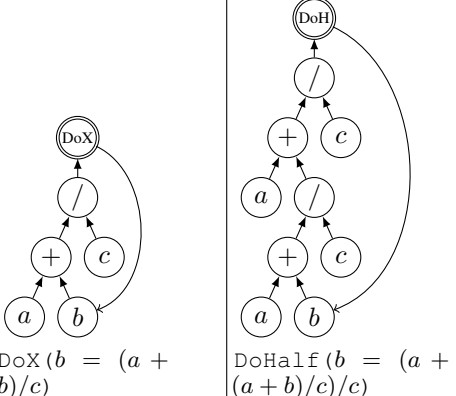

Figure 7: Adding loop constructs creates cycles in the program graph.

