# OpenReview forum: "Learning Axioms to Compute Verifiable Symbolic Expression Equivalence Proofs Using Graph-to-Sequence Networks"
_ICLR.cc/2021/Conference — Reject_

### Official Review · AnonReviewer2 · 2020-10-15
**Fine but not interesting enough for ICLR**

**Rating:** 4
**Confidence:** 3

**Review:**

The authors introduce a new synthetic dataset of equational proofs over the basic axioms of linear algebra.  The dataset consists of triples `(t1, t2, rewrites)` where `rewrites` is a sequence of rewrite instructions that can transform `t1` into `t2`, though some of their models emit one rewrite at a time and observe the result of applying the rewrite instruction to `t1`.  They develop a GNN for the task called pe-graph2axiom and show that it beats two baselines.  Finally, they show that their trained system can solve 15 problems from two Khan Academy modules that can be expressed in their fragment.  The appendix refers to supplementary material including code and data, but no supplementary material was submitted.

It is a fine paper, but ultimately I do not think the work is sufficiently interesting to merit publication at ICLR.  As far as I can tell, the entire synthetic dataset is easily decidable by traditional approaches, e.g. by simplifying and distributing modulo AC then checking for syntactic equality.  Their pe-graph2axiom model also seems relatively standard for theorem proving domains, and their Khan Academy validation set of 15 problems seems rather meager.

Other miscellaneous comments:
- The novelty claim in the abstract seems overstated. It is common to use GNNs for axiomatic theorem proving.
- The second sentence of the intro is unclear: "However, their stochastic nature tends to hinder their ability to learn representations useful for manipulating symbolic information." I don't think the word "stochastic" is being used correctly here.
- Later in the intro: "[the authors' new method] can deterministically prove equivalence". I find the intended meaning of the word "deterministically" unclear in this context.
- The "no false positives" point is made several times in the introduction, even though most ML-for-theorem-proving papers have operated in formal environments where this property holds.
- Page 2: I think `P1 ≡ A1(a, P1)` should be `P2 ≡ A1(a, P1)`
- Page 4: Why is it necessary for datapoints not to share `P1`s?
- I do not see discussion of the "Graph-to-sequence w/o attention" baseline presented in Table 3. What are the exact differences between this architecture and the pe-graph2axiom model? The question is particularly relevant since it performs almost as well.
- It seems odd that they do not compare against transformers.
- Page 8: "hypermeter" -> "hyperparameter"
- I personally did not find the "WholeProof" approach to add much to the paper

---

### Official Review · AnonReviewer3 · 2020-10-27
**Nice work, need a bit clarification about contributions.**

**Rating:** 5
**Confidence:** 5

**Review:**

Summary: This paper proposes a synthetic dataset of algebraic expressions with various kinds of symbols (e.g. scalars, vectors, matrices), and applies graph-to-sequence networks (with attentions) for predicting a sequence of rewrite rules (i.e. axioms) as an equivalent proof between two expressions. The prediction can be validated by a simple checker so that any false positives can be eliminated.

I like this work in general but hesitate to recommend acceptance before a few concerns can be addressed.

Detailed comments:
- Quality: The authors present a comprehensive discussion of related work. The dataset generation is carefully designed and very systematic. However, comparison with baseline approaches seems arbitrary. It is not clear what data is used for the mini ablation study shown in Table 3. The claim of "generate axiomatic proofs of equivalence between program pairs" is a bit over-claimed and misleading, as programs are way complicated than algebraic expressions. Similarly, "scalars, vectors and matrices" are mere symbols (associated with different kinds).

- Clarity: The overall writing is clear. The last part of section 5, i.e. the discussion about "Incremental versus non-incremental  sequence production" is quite confusing. Why path encoding is needed? How is it related to the "incremental/non-incremental" thing?

- Originality: Graph-to-sequence networks were proposed by Beck et al. 2018; leveraging type information with graph neural networks is not new as well (see [1]).

- Significance: This paper investigates a very important research topic, i.e. deep learning for symbolic reasoning, and proposes an interesting dataset, which could motivate many future works in this direction.

Questions:

Q0: How is the model trained? Particularly, how to provide (intermediate) supervision? Introducing intermediate programs makes AxiomStep5 and AxiomStep10 essentially "AxiomStep1" in training, i.e. "single-step prediction", is that right?  That means multiple-step prediction (with beam search) only happens in testing.

Q1: the authors argue "ensures correctness for all non-equivalent programs input", "no sequence S produced can be verified correct: true negatives are trivially detected.", which I find very hard to understand.  Correct me if I am wrong here. It is obvious that false positive (i.e. a sequence of rewrites is invalid) can be easily checked. But, if the model fails to find a sequence of rewrites, we are not sure if the input pair of _expressions_ are equivalent or not, how can we draw the conclusion that there is no such sequence?

Q2: given an input pair, do you limit the maximum number of generation steps? Say the ground truth takes 3 steps, the model may find a sequence of rewrites in 7 steps?  Do you count that as a success or failure?

Q3: regarding embeddings, do different symbols have different initial embedding? In NLP tasks, assigning the same token with the same embedding is less problematic.  Does the same symbol in different algebraic expressions have the same initial embedding?

Q4: After intermediate program generation, P1 is rewritten to P'. Will embeddings of P1 be reused in P' in the next step prediction? Or they may be simply reset and another single-step prediction is invoked?

Q5: How quantifiers are supported in axioms? Will "(a+b+c) - (a+b+c)" be reduced (or canceled) to 0 in one step? how many steps it might take?

Q6: Can the system prove equivalence between "1 + 1" and "2", or a slightly more complicated example, "a + a" and "2a"?

Minor typo:
Page 5, "production rulse" => "production rules"

[1] LambdaNet: Probabilistic Type Inference using Graph Neural Networks, ICLR 2020

---

### Official Review · AnonReviewer1 · 2020-10-28
**Detecting equivalent programs with rewriting**

**Rating:** 6
**Confidence:** 3

**Review:**

The authors present an algorithm for proving equivalence between algebraic expressions. Their solution relies on a graph-to-sequence neural network that generates the rewrite strategy for translating one program to the other. I think it is very positive that the authors implemented it in OpenNMP-py.

Comments
* On page (2) the claim (ii) is not novel in the sense that it produces a sequence of the proof, but because it uses the OpenNMT-py framework. Framing the theorem proving problem in a translation framework is very practical and useful.

* In page2 “...It can deterministically prove equivalence, entirely avoids false positives, and quickly invalidates incorrect answers produced by the network (no deterministic answer is provided in this case, only a probability of non-equivalence). “ Most of the systems provide this guarantee. It is always easy to verify the proof deterministically and check if it is valid. I don’t understand the emphasis on that fact

* I think it would have been useful if the authors had characterized their method as self-supervised. This is actually the way they generated data. Adopting this formalism it would make it much easier to communicate their writing easier to other research areas

* It would make a lot of sense to use the proposed technique for compiler rewrites in compiler optimization. I think there are other competing methods that are much more successful in theorem proving. This one seems to be more appropriate for compiler optimization.


* One of the details that is hidden in the appendix is that the algorithm is not invariant to the name of the variables. The algorithm recognizes only 4 variable names. For that reason would be curious to see how the algorithm behaves in the case of trying to reprove the same theorem it was trained on but with different variable names. Also, the authors should give a direction of how they can solve this limitation.


* One complaint that I have is that the appendix is unreasonably big (20 pages). I noticed that there is a lot of redundancy, as a lot of the same information is unnecessarily repeated in the appendix


* Figure 1 is very confusing. I struggle to understand how it maps to the description of the algorithm. I think the paper needs more graphics, given the complexity of the algorithm


* I don’t understand the use of the Graph Neural Network. Is it used to model the AST of the program and generate an embedding for the LSTM?


* The related work section is sufficient


* On the positive side of the paper, I liked the comparative study between incremental and non-incremental proof.


* The ability to generalize on longer proofs is definitely on the plus side of the paper

In general, I have a positive opinion about the paper, although it could have written in a more clear way.

---

### Official Review · AnonReviewer4 · 2020-10-28
**Though a valuable problem to solve, paper does not make a strong enough case for its proposed solution.**

**Rating:** 4
**Confidence:** 4

**Review:**

Summary:
This paper proposes a model for verifying semantic equivalence  between symbolic linear algebra expressions. Expressions are represented by trees and equivalence is proven by a sequence of axioms applied to the first expression. The proposed model encodes the expression/program trees as nodes on a graph connected by edges representing one of a set of axioms being applied to one of the elements in the first expression to yield a node in the second expression. The output of the model is a path, a sequence of edges, on this constructed graph that correspond to a sequence of axioms applied to the first expression to arrive at the second.

Strengths:
* The paper investigates an interesting problem of provably-correct equivalence proof generation for symbolic expressions.
* Results on incremental versus comprehensive supervision over the entire proof sequence motivate the training method proposed by the work.

Weaknesses:
* It’s not clear how this work fits into the context of the field and experiments are only conducted as ablation studies on the architecture choices and training regimes, with no other existing methods for expression rewriting or proof generation considered.
* The system is trained and tested on expressions that are very few rewrite steps away. While datasets were generated for up to 5 and 10 rewrite steps, even the 10-step dataset contains mostly pairs that have fewer (1-5) rewrite steps. In any case, 5 and 10 steps are very few and claims of robustness and generalization are weak with such few steps in the required proofs and such a large overlap in the steps required for pains in both datasets.
* The system relies strongly on training one step at a time by requiring samples generated for every intermediate step in the proof of equivalence between two expressions.
* There is no motivation for using a deep model over another path-finding method on the constructed graph that connects the input programs.

Recommendation:
I recommend against acceptance of the paper in its current form. The idea of GNN based path-finding for expression/program equivalence proving is promising, but the methods are not clearly and rigorously explained and the experimental settings are too weak to support the claims of generalizability and applicability to realistic scenarios.

Update: Thank you to the authors for clarifying the points brought up in reviews. I acknowledge that I have reviewed their responses.

Questions:
1. The development of the equivalence proofs on page 3 is very unclear. What does it mean to say “$x \in v_i$”? This notation is not well explained. What is $x$ and what does it mean for $x$ to be _in_ $v_i$?
2. The text says that models were “run” twice, does this mean trained twice with different initializations? If so, it would be more informative to report average performance rather than the one from the best random seed. If this is not the case, could you please clarify the training set-up?
3. What are the evaluation metrics used and reported? What measure was used for the validation score that determined the best-performing model?

Minor comments:
* It is best practice to define acronyms the first time they are used. This will avoid confusion, especially for readers from a wider machine learning audience. In particular, GNN, GGNN and AST are not defined
* There are many cases where citations should be within parentheses, please check for these. If the work is not referenced as part of the text of the sentence, it should be in parentheses for clarity.

Typos:
Pg 5, “production rulse" should read “production rules”
Pg 5, there is a sentence that begins with “E.g.” which should be replaced with the grammatical “For example,”. Similarly, on page 5, the sentence that begins “Fig. 1” should read “Figure 1”.

---

### Decision · Program_Chairs · 2021-01-07
**Final Decision**

**Decision:**

Reject

**Comment:**

I think this is a very promising paper, but the work is not ready for publication.

The most significant concern shared by several reviewers is the insufficient evaluation. For example, the work is not compared with more traditional approaches to equivalence checking or any other baselines beyond ablations of the proposed method. Given that this is not the first paper to propose the use of deep learning to search for proofs, it seems important to compare to alternative methods. There is also a misalignment between the claims of novelty and the evaluation. For example, section 4 cites the novel approach to generating data as key to this approach, but the evaluation does not really address this claim. On the positive side, I was impressed with the ability to search for proofs of length 10 given the large branching factor, and I thought the results were promising.

The authors should also consider some of the concerns with presentation raised by the reviewers.